# Phosphatidylserine exposure mediated by ABC transporter activates the integrin signaling pathway promoting axon regeneration

Naoki Hisamoto[1], Anna Tsuge[1], Strahil Iv. Pastuhov[1], Tatsuhiro Shimizu[1], Hiroshi Hanafusa[1] & Kunihiro Matsumoto[1]

Following axon injury, a cascade of signaling events is triggered to initiate axon regeneration. However, the mechanisms regulating axon regeneration are not well understood at present. In *Caenorhabditis elegans*, axon regeneration utilizes many of the components involved in phagocytosis, including integrin and Rac GTPase. Here, we identify the transthyretin (TTR)-like protein TTR-11 as a component functioning in axon regeneration upstream of integrin. We show that TTR-11 binds to both the extracellular domain of integrin-α and phosphatidylserine (PS). Axon injury induces the accumulation of PS around the injured axons in a manner dependent on TTR-11, the ABC transporter CED-7, and the caspase CED-3. Furthermore, we demonstrate that CED-3 activates CED-7 during axon regeneration. Thus, TTR-11 functions to link the PS injury signal to activation of the integrin pathway, which then initiates axon regeneration.

---

[1] Division of Biological Science, Graduate school of Science, Nagoya University, Chikusa-ku, Nagoya 464-8602, Japan. These authors contributed equally: Naoki Hisamoto, Anna Tsuge, Strahil Iv. Pastuhov. Correspondence and requests for materials should be addressed to N.H. (email: i45556a@cc.nagoya-u.ac.jp) or to K.M. (email: g44177a@nucc.cc.nagoya-u.ac.jp)

Some neurons possess the fundamental and conserved ability to regenerate their axons after injury. This ability is regulated by extracellular factors in the local environment in concert with the neuron's intrinsic machinery controlling growth cone formation and extension. In adult mammals, axons of the peripheral nervous system regenerate relatively efficiently, whereas axons of the central nervous system regenerate poorly[1,2]. This difference has been attributed to both extrinsic signals provided by the inhibitory glial environment and intrinsic axon growth capabilities[2]. The latter are believed to be primary in determining regenerative success, and thus have been the focus of considerable effort to identify potential therapeutic targets that might promote nerve regeneration. Nevertheless, at present our understanding of these intrinsic signaling mechanisms are limited.

The nematode *Caenorhabditis elegans* is an attractive model to dissect the mechanism of axonal regeneration[3]. Genetic studies in *C. elegans* have identified novel signaling pathways involved in this process[4] and have provided invaluable insights into the signaling networks that regulate axon regeneration[5]. One of such signaling pathway is the c-Jun N-terminal kinase (JNK) MAP kinase (MAPK) pathway[6,7]. MAPK signaling pathways respond to various extracellular stimuli and function to regulate cell proliferation, differentiation, regeneration, response to stress, and apoptosis[8,9]. The JNK pathway in *C. elegans* is comprised of MLK-1 MAPKKK, MEK-1 MAPKK, and KGB-1 JNK[6], within which activation of the upstream MAPKKK is critical in determining signal specificity[10]. The protein kinase MAX-2, related to yeast Ste20, phosphorylates and activates MLK-1. Upstream of MAX-2 is the integrin-α subunit INA-1, which signals through the guanine nucleotide exchange factor complex, CED-2, CED-5, and CED-12. This complex in turn activates the Rac-type GTPase CED-10, which initiates axon regeneration[11]. Interestingly, the INA-1–CED-10 signaling pathway is also involved in the phagocytosis of apoptotic cells during development[12], thus this signaling module regulates both the engulfment of dying cells and axon regeneration.

During apoptosis, apoptotic cells display phosphatidylserine (PS) on their surface, which functions as an eat-me signal. Phagocytes recognize the PS signal either directly through engulfment receptors or indirectly through linker molecules that act between apoptotic cells and phagocytes[13]. Mammalian integrins can bind to PS on apoptotic cells indirectly via linker molecules such as the secreted MFG-E8 (ref. [14]). However, *C. elegans* apparently does not contain a homolog of MFG-E8 that can be easily identified by sequence analysis. Previous studies have shown that the transthyretin (TTR)-like protein, TTR-52, in *C. elegans* functions as a linker molecule that bridges PS externalized on the apoptotic cell surface and the CED-1 receptor on the engulfing cell[15]. A recent study also has demonstrated that PS is exposed as a result of axon severing and promotes axonal fusion in PLM neurons through the TTR-52-dependent signaling pathway[16]. Our previous results suggested that the INA-1–CED-10 pathway regulating this engulfment of apoptotic cells has been evolutionarily co-opted for the regulation of axon regeneration[11]. These findings raised the possibility that externalized PS generated by axon severing is recognized by INA-1 indirectly via a linker molecule to activate the INA-1–CED-10 pathway.

In this study, we identify a TTR-like protein, TTR-11, as a component functioning upstream of INA-1 in axon regeneration. The TTR-11 protein binds to the extracellular domain of INA-1 and to PS. Axon injury induces the accumulation of PS around the injured axons in a manner dependent on TTR-11, the ABC transporter CED-7, and the caspase CED-3. Our results support a model in which TTR-11 mediates the recognition of injured axons by cross-linking the PS signal with integrin, suggesting that

PS exposure acts as the initial signal that directs the injured axon to initiate regeneration.

## Results

**TTR-11 is involved in axon regeneration.** Recent genetic studies in *C. elegans* have revealed that axon regeneration is regulated by the JNK MAPK pathway[6]. The JNK cascade can be inactivated by the MAPK phosphatase VHP-1, and *vhp-1*-null mutants cause an arrest in developmental growth at the early larval stage due to hyperactivation of the JNK pathway (Supplementary Fig. 1)[17]. Indeed, loss-of-function mutations in the *mlk-1*, *mek-1*, or *kgb-1* gene can suppress this larval arrest phenotype[17]. We previously isolated suppressors of *vhp-1* lethality (*svh* genes) by a genome-wide RNA interference (RNAi) screen and found that these function in the JNK pathway (Supplementary Fig. 1)[18]. We isolated 92 of these *svh* RNAi clones. In this study, we investigated the roles of the *svh-13*/F46B3.3 and F46B3.18 RNAi clones in axon regeneration.

*svh-13* RNAi is expected to target both the *ttr-11* and *ttr-57* genes (Fig. 1a). To determine if either gene is involved in axon regeneration, we characterized regeneration in *ttr-57(tm6877)*-null mutant animals (Fig. 1a), as well as a *ttr-11(km64)* mutant we generated using the CRIPSR/Cas9 system (Fig. 1a) (see Methods). We assayed the regrowth of laser-severed axons in GABA-releasing D-type motor neurons, which extend their axons from the ventral side to the dorsal side in the animal body (Fig. 1b)[3,19]. In wild-type animals at the L4 stage, axons severed by laser initiated regeneration within 24 h (Fig. 1b, c and Supplementary Table 1), while in *ttr-11(km64)* mutants, the frequency of axon regeneration was reduced (Fig. 1b, c and Supplementary Table 1). In contrast, *ttr-57* mutations affected neither axon regeneration itself nor the defect in regeneration observed in *ttr-11(km64)* mutants (Fig. 1c, Supplementary Fig. 2, and Supplementary Table 1). Furthermore, overexpression of *ttr-57* did not influence the *ttr-11* defect in regeneration (Supplementary Fig. 2 and Supplementary Table 1). These results indicate that TTR-11, but not TTR-57, is involved in axon regeneration after laser axotomy.

To confirm that the axon regeneration defect was indeed caused by the *ttr-11* mutation, we made a P*ttr-11::ttr-11* transgene that contains approximately 2.4 kb of genomic DNA, including the entire *ttr-11* coding region and its promoter. Introduction of P*ttr-11::ttr-11* as an extrachromosomal array rescued the regeneration defect of *ttr-11(km64)* mutants (Fig. 1c and Supplementary Table 1). We also found that expression of *ttr-11* in sensory neurons by the *mec-7* promoter, but not in D-type neurons by the *unc-25* promoter, rescued the axon regeneration defect of D motor neurons in *ttr-11(km64)* mutants (Fig. 1c, Supplementary Fig. 2, and Supplementary Table 1). These results thus indicate that TTR-11 can act non-autonomously in axonal regeneration.

To determine the expression pattern of the *ttr-11* gene, we constructed a transgene P*ttr-11::nls::venus*, which expresses the fluorescent protein VENUS fused to a nuclear localization signal (NLS) under the control of the *ttr-11* promoter. In L1 stage larvae, animals carrying P*ttr-11::nls::venus* exhibited the expression of VENUS in HSN neurons, excretory gland cells, hypodermal hyp10 cells, and DVA neurons, but not in D-type motor neurons (Supplementary Fig. 3). However, this expression pattern was not detected in L4 stage animals and expression of VENUS was not visible in D motor neurons after axon injury (Supplementary Fig. 4). Since the native introns and exons of the *ttr-11* gene might be critical to tissue-specific expression, we also examined the expression pattern of *ttr-11* using the intact genomic version of the gene, P*ttr-11::ttr-11::gfp*. GFP expression was still not observed around D motor neurons in L4 stage animals after axon injury (Supplementary Fig. 5).

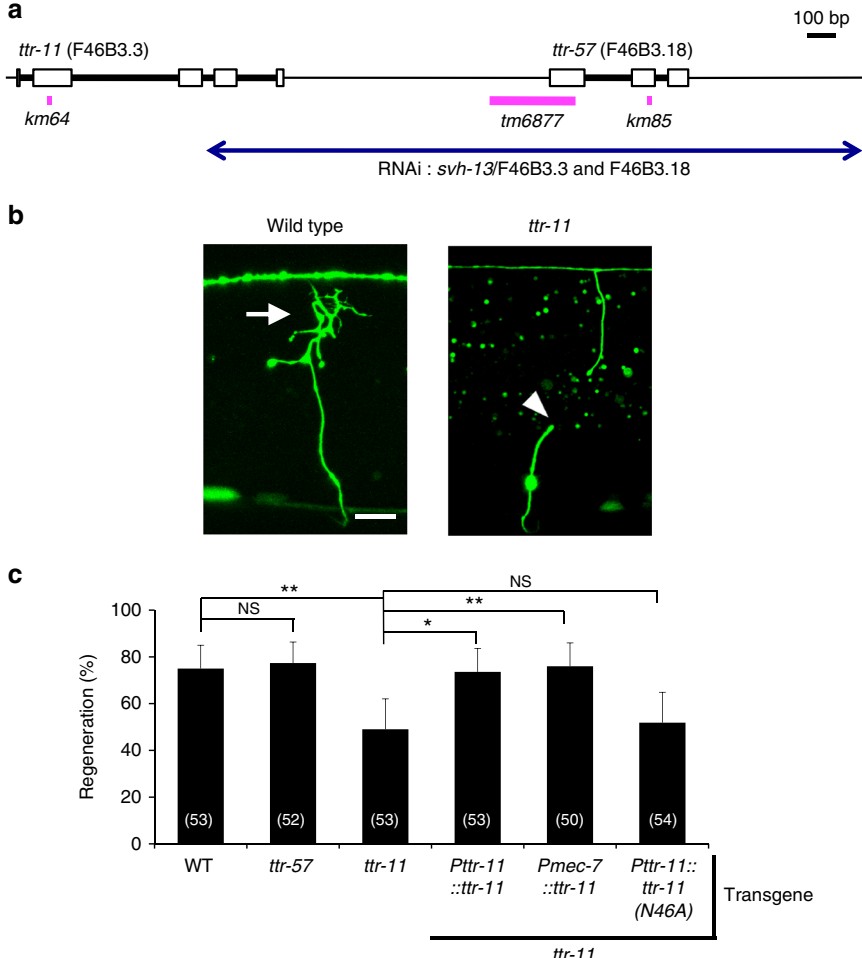

**Fig. 1** TTR-11 is required for efficient axon regeneration. **a** Genomic structures of the *ttr-11* and *ttr-57* genes and the *svh-13*/F46B3.3 and F46B3.18 RNAi clone. The *svh-13* RNAi knocks down both the *ttr-11* and *ttr-57* genes. Boxes and thick lines indicate exons and introns, respectively. The blue line indicates the region targeted by the *svh-13* RNAi. The magenta bold lines underneath indicate the extent of the deleted region in each deletion mutant. **b** Positive and negative regeneration in D-type motor neurons. The positive and negative regeneration examples are from a wild-type animal and a *ttr-11* mutant, respectively, 24 h after laser surgery in the L4 stage. In wild-type animals, a severed axon has regenerated a growth cone (arrow) and ~75% of the cut axons are scored as regenerating. In *ttr-11* mutants, proximal ends of axons often failed to regenerate (arrowhead). Scale bar = 10 μm. **c** Regeneration of D-type motor neurons. Percentages of axons that initiated regeneration 24 h after laser surgery in the L4 stage are shown. The number (*n*) of axons examined are shown. Error bars indicate 95% confidence intervals (CIs). *$P < 0.05$, **$P < 0.01$ as determined by Fisher's exact test. NS: not significant

**TTR-11 binds to both PS and integrin.** Mammalian MFG-E8 binds extracellularly to PS via its C-terminal region (Fig. 2a)[14]. The TTR-11 protein has a secretion signal sequence and shares limited sequence similarity to TTR (Supplementary Fig. 6). Interestingly, it has been shown that *C. elegans* TTR-52 binds to PS[15,20]. Therefore, we examined the possibility that TTR-11 also binds to PS. We purified FLAG-tagged TTR-11 (TTR-11:: FLAG) proteins from mammalian HEK293 (human embryonic kidney 293) cells and tested their ability to bind to phospholipids. TTR-11 showed strong binding to PS and phosphatidic acid (PA) but not to other phospholipids (Fig. 2b). TTR-11 also exhibited weak binding to phosphatidylinositol-4-phosphate, phosphatidylinositol-4,5-diphosphate, and phosphatidylinositol-3,4,5-triphosphate (Fig. 2b).

Since MFG-E8 serves as a linker molecule that recognizes apoptotic cells by bridging PS and integrin (Fig. 2a)[14], we examined whether TTR-11 associates with integrin-α INA-1. We expressed the GFP-tagged extracellular domain of INA-1 (INA-1-ECD::GFP) and TTR-11::FLAG in HEK293 cells separately. Cell lysates were prepared from each cell and mixed

in vitro and immunoprecipitated with anti-FLAG antibody. We found that TTR-11::FLAG co-precipitated INA-1-ECD::GFP (Fig. 2c). Similarly, immunoprecipitation of INA-1-ECD with anti-GFP antibody co-immunoprecipitated TTR-11::FLAG (Supplementary Fig. 7). These results suggest that TTR-11 can act as a linker molecule between PS and the INA-1 receptor.

The Asp-51 residue in TTR-52 is essential for apoptotic cell engulfment[20]. The corresponding site (Asn-46) is conserved in TTR-11 (Supplementary Fig. 6). To examine whether the Asn-46 site in TTR-11 is required for axon regeneration, we generated a mutant form of TTR-11 [TTR-11(N46A)], in which Asn-46 is mutated to alanine (Supplementary Fig. 6). We found that the N46A point mutation was unable to rescue the axon regeneration-defective phenotype of *ttr-11(km64)* mutants (Fig. 1c and Supplementary Table 1). Furthermore, the in vitro association between the TTR-11(N46A)::FLAG-mutated form and INA-1-ECD::GFP was significantly weaker (Fig. 2c and Supplementary Fig. 7). These results suggest that the Asn-46 site in TTR-11 is important for binding to INA-1.

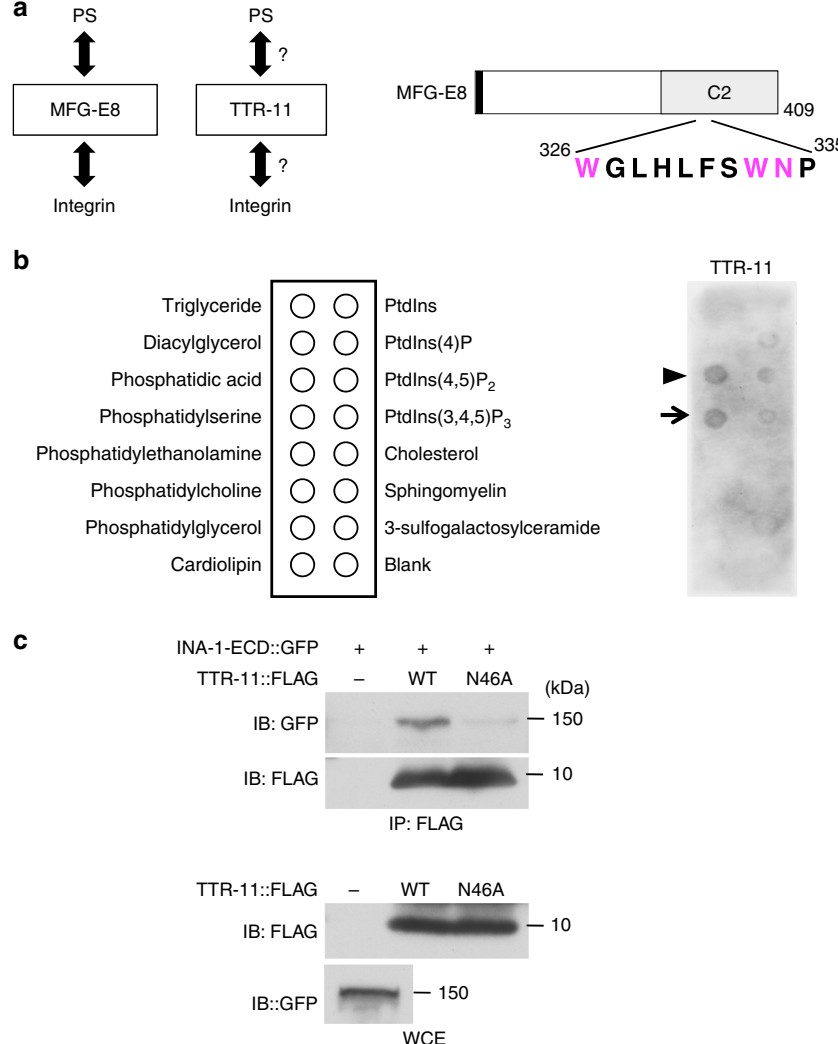

**Fig. 2** TTR-11 associates with both phosphatidylserine (PS) and integrin. **a** Comparison of the functions between MFG-E8 and TTR-11. Schematic diagram of MFG-E8 is shown in the right part. The black box indicates the predicted secretion signal. A C2 domain (C2) is also indicated. Three amino acid residues within the C2 domain (Trp-326, Trp-333, and Asn-334) essential for its PS binding are indicated in magenta. **b** Binding of TTR-11 with phospholipids. Affinity-purified TTR-11::FLAG bound to phospholipids on a membrane lipid strip. An arrow and an arrowhead indicate phosphatidylserine and phosphatidic acid, respectively. **c** Interaction of TTR-11 with the INA-1 extracellular domain in vitro. HEK293 cells were transfected with plasmids encoding INA-1-ECD::GFP and TTR-11::FLAG (wild type and N46A) separately. Cell extracts were prepared from each cell and mixed in vitro. Complex formation was detected by immunoprecipitation (IP) with an anti-FLAG antibody, followed by immunoblotting (IB) with an anti-GFP antibody. Whole-cell extracts (WCEs) were analyzed by immunoblotting

**TTR-11 functions upstream of the integrin–CED-10 pathway.** Next, we examined whether TTR-11 and INA-1 function in the same pathway in axonal regeneration (Fig. 3a). We found that the regenerative defect in *ttr-11(km64); ina-1(gm39)* double mutants was not stronger than that observed in the individual mutants (Fig. 3b and Supplementary Table 1), suggesting that INA-1 and TTR-11 act in the same pathway. In axon regeneration, INA-1 functions upstream of the Rac-type GTPase CED-10 (Fig. 3a)[11]. To address whether *ttr-11* and *ced-10* function in the same pathway, we constructed *ttr-11(km64); ced-10(n3246)* double mutants. The double mutants did not show any enhanced defect in axon regeneration compared to the single *ttr-11(km64)* or *ced-10(n3246)* mutants (Fig. 3b and Supplementary Table 1), suggesting that TTR-11 and CED-10 also function in the same pathway. If TTR-11 functions in axon regeneration upstream of CED-10, we would expect that a constitutively active mutation of the *ced-10* gene would suppress the *ttr-11* phenotype. We expressed CED-10 (G12V), a mutant CED-10 locked in the

active, GTP-bound form, in D-type motor neuron by the *unc-25* promoter and found that it indeed suppressed the *ttr-11* defect in axon regeneration (Fig. 3b and Supplementary Table 1). In contrast, similar expression of CED-10(T17N), a GDP-bound inactive form, did not suppress the *ttr-11* defect (Fig. 3b and Supplementary Table 1). These results suggest that TTR-11 functions upstream of the INA-1–CED-10 pathway regulating axon regeneration.

**Axon injury induces PS accumulation around injured axons.** We next analyzed the dynamics of surface-exposed PS expression after axonal injury using a fusion between GFP and MFG-E8-C2, which is secreted and binds selectively to PS, as a fluorescent PS biosensor (MFG-E8-C2::GFP) (Fig. 2a)[21]. The *mfg-e8-c2::gfp* fusion gene was expressed using a heat-shock promoter and we monitored the dynamics of MFG-E8-C2::GFP localization before and after D-type neuron axotomy. Before axon injury, there was

**a**

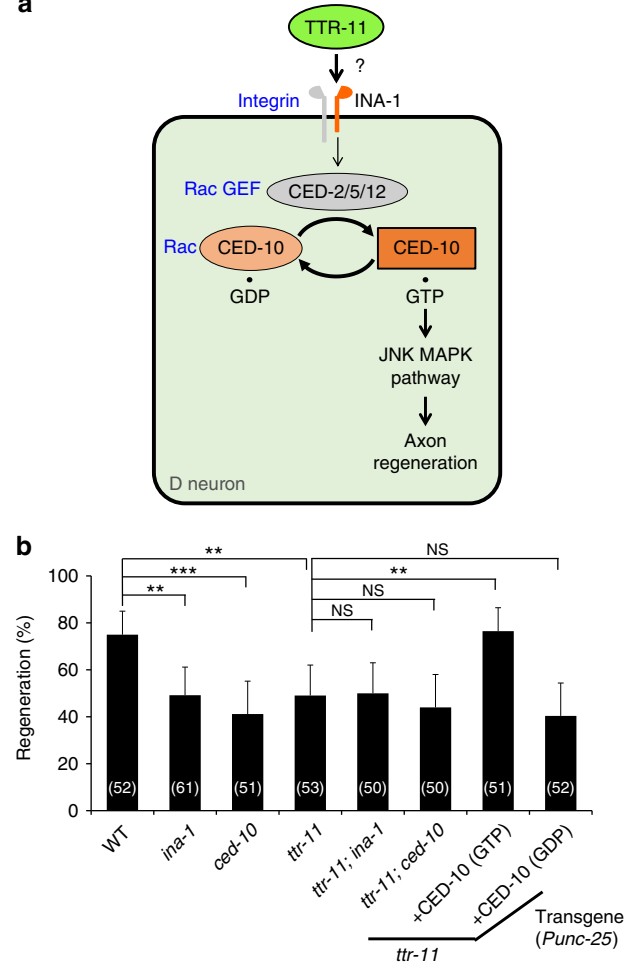

**Fig. 3** TTR-11 functions in the INA-1–CED-10 pathway. **a** Schematic diagram of the INA-1–CED-10 pathway regulating axon regeneration. GEF GDP-GTP exchange factor, CED-10(GTP) GTP-binding form of CED-10, and CED-10 (GDP) GDP-binding form of CED-10. **b** Percentages of axons that initiated regeneration 24 h after laser surgery in the L4 stage. The number (*n*) of axons examined are shown. Error bars indicate 95% CI. **P < 0.01, ***P < 0.001 as determined by Fisher's exact test. NS: not significant

no particular localization of MFG-E8-C2::GFP around the D motor neurons (Fig. 4a, b and Supplementary Fig. 8). However, we observed localization of MFG-E8-C2::GFP around the injured D neurons following axotomy (Fig. 4a, b and Supplementary Fig. 8). At 10 min after axotomy, MFG-E8-C2::GFP expression was observed between the proximal and distal axon segments (Supplementary Fig. 9a). This localization of MFG-E8-C2::GFP began diffusing by 2 h after surgery. Thus, axon injury induces the transient localization of MFG-E8-C2::GFP around D motor neurons. Furthermore, we found that when MFG-E8-C2::GFP was expressed in D motor neurons by the *unc-25* promoter, D neuron axotomy induced MFG-E8-C2::GFP localization around the injured D neurons (Supplementary Fig. 9b). Similar localization patterns were also observed when we expressed another PS-binding protein, Annexin V::GFP (AnxV::GFP), by a heat-shock promoter (Supplementary Fig. 9c).

Three amino acid residues (Trp-326, Trp-333, and Asn-334) in MFG-E8-C2 are essential for binding to PS (Fig. 2a)[21]. When we expressed MFG-E8-C2(AAA)::GFP, a variant that does not bind PS due to the replacement of these three residues to alanines[21], no increase in fluorescence localization was observed following axon injury (Fig. 4a, b and Supplementary Fig. 8). This confirms that

localization of MFG-E8-C2::GFP reflects the distribution of PS and indicates that PS is exposed around the injured axons after injury. We also found that expression of MFG-E8-C2::GFP inhibited axon regeneration (Fig. 4c), which might be due to its ability to sequester exposed PS. By contrast, expression of MFG-E8-C2(AAA)::GFP did not appear to affect regeneration (Fig. 4c). Thus, the interaction of TTR-11 with PS is likely important for axon regeneration.

What regulates PS exposure in response to axon injury? A flippase (out-to-in translocation) normally confines PS to the inner leaflet of the plasma membrane[22]. In mammals, apoptosis activates a scramblase, resulting in the exposition of PS on the cell surface. Recently, two membrane proteins, transmembrane protein 16F (TMEM16F) and Xk-related protein 8 (Xkr8), have been identified as factors responsible for phospholipid scrambling in mammalian membranes[22]. *Caenorhabditis elegans* contains homologs of mammalian TMEM16F and Xkr8, which are the *anoh-1* and *ced-8* genes, respectively (Fig. 5a)[23,24]. However, we found that loss-of-function mutations in either the *anoh-1* or *ced-8* gene and the *anoh-1; ced-8* double mutation had no significant effect on axon regeneration (Fig. 5b and Supplementary Table 1). Thus, ANOH-1 and CED-8 do not seem to be involved in PS exposure during axon regeneration.

The mammalian ABC transporter A1 (ABCA1) translocates PS from the inner to the outer leaflet[25]. The *C. elegans ced-7* gene encodes a member of ABC transporter family (Fig. 5a)[26]. Interestingly, we found that the *ced-7(n2094)* mutation decreased the frequency of axon regeneration (Fig. 5b and Supplementary Table 1). We therefore examined whether loss of *ced-7* affects PS exposed on the injured axon during axon regeneration by analyzing the localization of surface-exposed PS using MFG-E8-C2::GFP. PS labeling around the injured neurons following axotomy was weaker in *ced-7(n2094)* mutants compared to wild-type animals (Fig. 5c, d and Supplementary Fig. 8), suggesting that CED-7 is important for inducing PS exposure on the injured axon. Maintenance of lipid asymmetry is accomplished by the type IV P-type ATPase family of proteins (P4-ATPases) that function as phospholipid flippases[22]. Since the *C. elegans* CHAT-1 protein acts as a chaperone of the P4-ATPase TAT-1, the *chat-1* mutation causes ectopic exposition of PS on the surface of living cells[27]. Consistent with this, we found that the *chat-1(ok1681)* mutation was able to suppress the axon regeneration defect of *ced-7(n2094)* mutants (Fig. 5b and Supplementary Table 1).

To examine whether *ced-7* and *ttr-11* function in the same pathway to regulate axon regeneration, we constructed *ced-7 (n2094); ttr-11(km64)* double mutants. The double mutants did not show any enhanced defect in axon regeneration compared to the single *ced-7(n2094)* mutant (Fig. 5e and Supplementary Table 1), suggesting that TTR-11 and CED-7 function in the same pathway. Furthermore, overexpression of the *ttr-11* gene and a constitutively active CED-10(G12V) mutant were each able to suppress the *ced-7* defect in axon regeneration (Fig. 5e, Supplementary Fig. 10, and Supplementary Table 1). These results support the possibility that CED-7 functions upstream of the TTR-11–CED-10 pathway in axon regeneration.

Next, we examined whether CED-7 can act cell autonomously by expressing the *ced-7* complementary DNA (cDNA) from the *unc-25* promoter in *ced-7(n2094)* mutants. Expression of *ced-7* in D motor neurons rescued the *ced-7* defect in axon regeneration (Fig. 5e and Supplementary Table 1). This result indicates that CED-7 acts in the damaged neuron to promote regeneration. We further examined whether *ced-7* expression outside the injured neuron can regulate axon regeneration. To test this possibility, the *mec-7* promoter was used to express the *ced-7* gene in touch neurons. Touch neuron axons run parallel to the body axis and cross perpendicularly to D-type neuron axons (Supplementary Fig. 11a). Expression of *ced-7* in

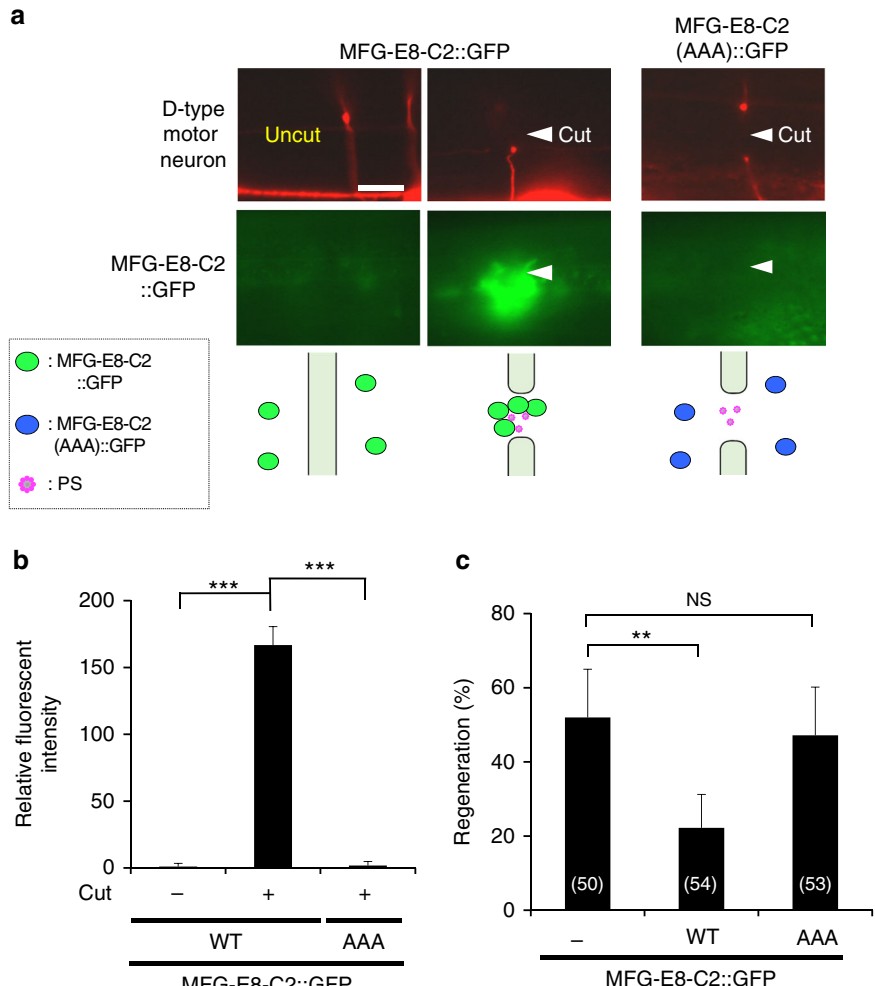

**Fig. 4** Axon injury induces PS exposure. **a** Localization of MFG-E8-C2::GFP after axon injury. Fluorescent images of severed axons in animals carrying *Phsp:: ss::mfg-e8-c2::gfp* or *Phsp::ss::mfg-e8-c2(AAA)::gfp* and *Punc-47::mcherry* are shown. D neurons are visualized by mCherry under control of the *unc-47* promoter. Images were taken just before (uncut) or 1 h after laser surgery (cut). Twenty animals were examined for each condition. All 20 observations showed similar patterns. Arrowheads indicate the sites of laser surgery. Schematic diagrams of GFP localization are shown in the lower part. Scale bar = 10 μm. **b** The relative fluorescent intensities of MFG-E8-C2::GFP around D neurons with (+) or without (−) laser surgery. Scores were calculated from images taken just before or 1 h after laser surgery. Quantification of MFG-E8-C2::GFP is described in Methods. Twenty animals were examined for each condition. Error bars indicate SEM. ***$P < 0.001$ as determined by unpaired $t$ test. **c** Percentages of axons that initiated regeneration 24 h after laser surgery in the L4 stage. Axotomy was performed after heat shock as described in Methods. The number ($n$) of axons examined are shown. Error bars indicate 95% CI. **$P < 0.01$ as determined by Fisher's exact test. NS: not significant

touch neurons did not rescue the *ced-7* defect in D-type motor neuron regeneration (Supplementary Fig. 11b and Supplementary Table 1). This result is consistent with the possibility that CED-7 functions cell autonomously. On the other hand, when both touch and D neurons were injured simultaneously in *ced-7(n2094)* mutants expressing *ced-7* in touch neurons, the regeneration defect of D neurons was suppressed (Supplementary Fig. 11b and Supplementary Table 1). These results suggest that CED-7 in the damaged touch neuron induces PS exposure, which acts on the damaged D neuron to regenerate through the TTR-11 signaling pathway.

During apoptosis, CED-7 mediates the release of PS from dying cells. In addition, TTR-52 could function as an extracellular PS carrier to facilitate further PS movement[28]. We next examined whether TTR-11 is involved in PS accumulation around the axon segments of D-type motor neurons following axotomy. We found that localization of MFG-E8-C2::GFP around the injured neurons following axon injury was lower in the *ttr-11* mutant vs. wild-type (Fig. 5c, d and Supplementary Fig. 7). This result suggests that TTR-11 also functions upstream to control PS accumulation after axon

injury. It is therefore possible that TTR-11 is required for PS exposure by acting as an extracellular PS carrier that facilitates PS movement. If so, we would expect TTR-11 to be localized around injured neurons following axon injury. However, we failed to detect GFP generated by a *Pttr-11::ttr-11::gfp* transgene after axon injury (Supplementary Fig. 5). This failure is probably due to the fact that fusion of GFP to either the C terminus or the N terminus after the signal sequence of *ttr-11* did not produce a functional fusion gene.

**Caspase CED-3 promotes axon regeneration by activating CED-7.** How is CED-7 activated during axon regeneration? The activity of mammalian ABCA1 is negatively regulated by a 40 amino acid C-terminal region[29]. Interestingly, the CED-7 C-terminal region contains the sequence DQXD, which is a potential site for proteolytic cleavage by caspase-3 (Fig. 6a)[30]. It has been reported that *C. elegans* caspase CED-3 is required for efficient axon regeneration[31], thus one hypothesis is that CED-7 is activated by a caspase that removes the C-terminal region. As

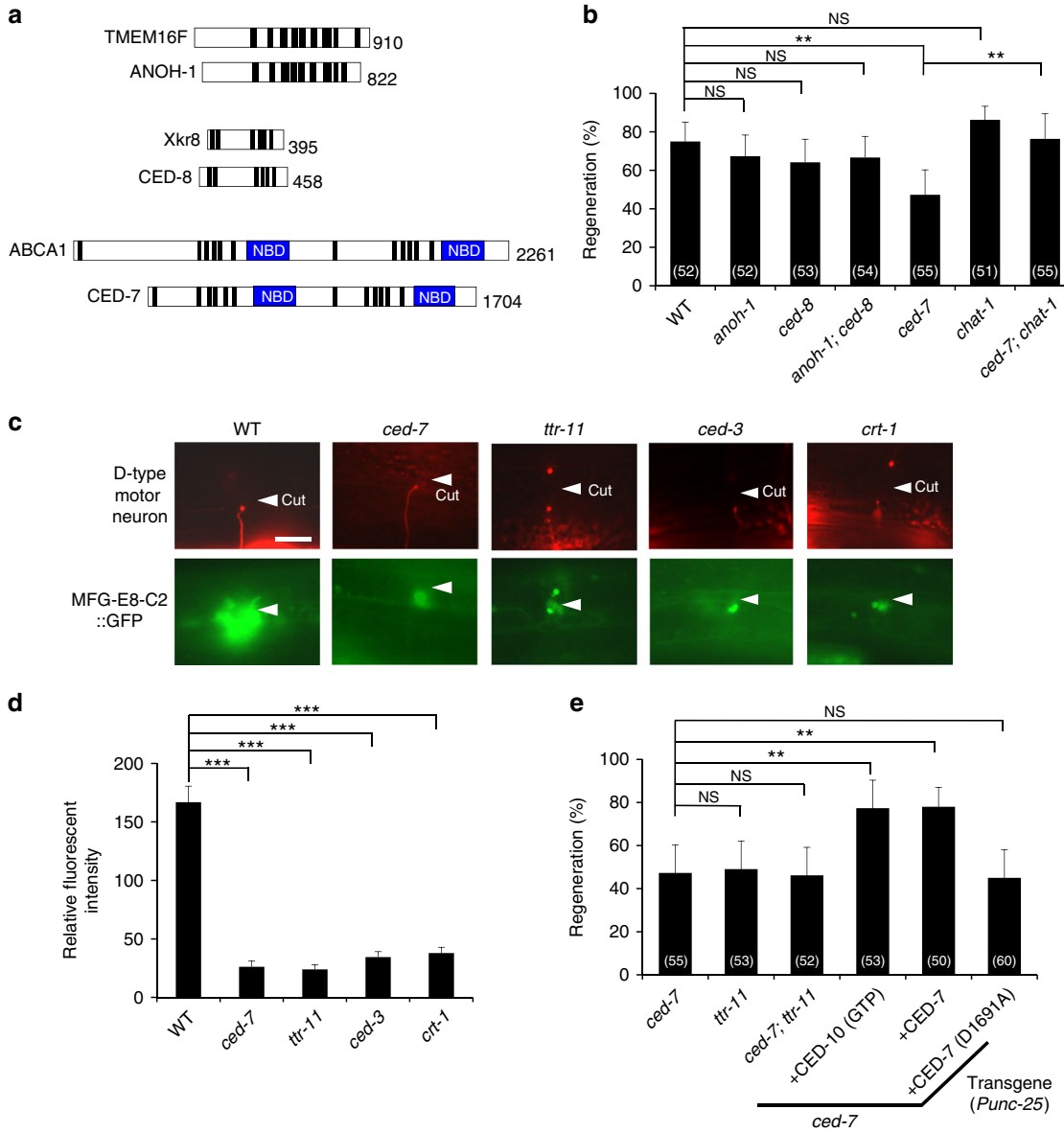

**Fig. 5** CED-7 is required for efficient axon regeneration. **a** Domain structures of *C. elegans* ANOH-1, CED-8, and CED-7. Schematic domain diagrams of *C. elegans* ANOH-1, CED-8, CED-7 and mammalian counterparts (TMEM16F, Xkr8, and ABCA1) are shown. Transmembrane domains and nucleotide-binding domains (NBD) are indicated by black and blue colors, respectively. **b**, **e** Percentages of axons that initiated regeneration 24 h after laser surgery in the L4 stage. The number (*n*) of axons examined are shown. Error bars indicate 95% CI. **P < 0.01 as determined by Fisher's exact test. NS: not significant. **c** Localization of MFG-E8-C2::GFP after axon injury. Fluorescent images of severed axons in animals carrying *Phsp::ss::mfg-e8-c2::gfp* and *Punc-47::mcherry* are shown. D neurons are visualized by mCherry under control of the *unc-47* promoter. Each image was taken at 1 h after laser surgery. Twenty animals were examined for each condition. All 20 observations showed similar patterns. Arrowheads indicate the sites of laser surgery. Scale bars = 10 μm. **d** The relative fluorescent intensities of GFP around D neurons with laser surgery. Scores were taken at 1 h after laser surgery. Quantification of MFG-E8-C2::GFP is described in Methods. Twenty animals were examined for each condition. Error bars indicate SEM. ***P < 0.001 as determined by unpaired *t* test

observed previously, we confirmed that *ced-3(ok2734)* mutants are significantly defective in axon regeneration following axotomy (Fig. 6b and Supplementary Table 1). To examine the relationship between *ced-3* and *ced-7* in axon regeneration, we compared the regenerative capacity of *ced-3(ok2734)* mutants, *ced-7(n2094)* mutants, and *ced-3(ok2734); ced-7(n2094)* double mutants. We found that the regeneration defects in the double mutants were similar to those in the single mutants (Fig. 6b and Supplementary Table 1), suggesting that CED-3 and CED-7 function in the same pathway to regulate axon regeneration. We then tested whether the abnormal PS distribution phenotype was also present in animals carrying the *ced-3(ok2734)* mutation. We found that PS

labeling around injured D-type motor neurons following axotomy was weaker in the *ced-3(ok2734)* mutation than wild-type (Fig. 5c, d and Supplementary Fig. 8).

Next, we genetically addressed the relationship between *ced-3* and *ced-7* in axon regeneration. We expressed the mutant CED-7 (D1691A), in which the Asp-1691 site in the DQXD motif is mutated to alanine (Fig. 6a), and found that it was unable to rescue the defect in axon regeneration observed in *ced-7(n2094)* mutants (Fig. 5e and Supplementary Table 1). This suggests that the caspase recognition site (D1691) in CED-7 is important for its function in axon regeneration. We also found that expression of a truncated form of CED-7 lacking its C-terminal region (CED-

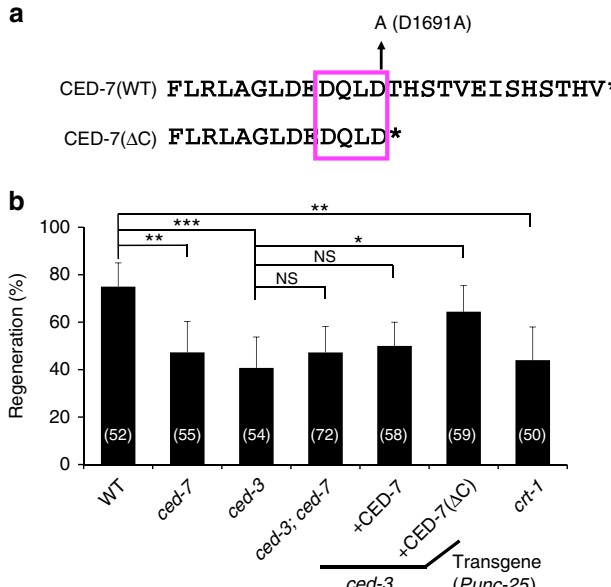

**Fig. 6** Caspase CED-3 acts upstream of CED-7 in axon regeneration. **a** Caspase recognition sequence of CED-7. The CED-3 recognition sequence, DQXD, in its C-terminal region is boxed with a magenta line. **b** Percentages of axons that initiated regeneration 24 h after laser surgery in the L4 stage. The number (*n*) of axons examined are shown. Error bars indicate 95% CI. *$P < 0.05$, **$P < 0.01$, and ***$P < 0.001$ as determined by Fisher's exact test. NS: not significant

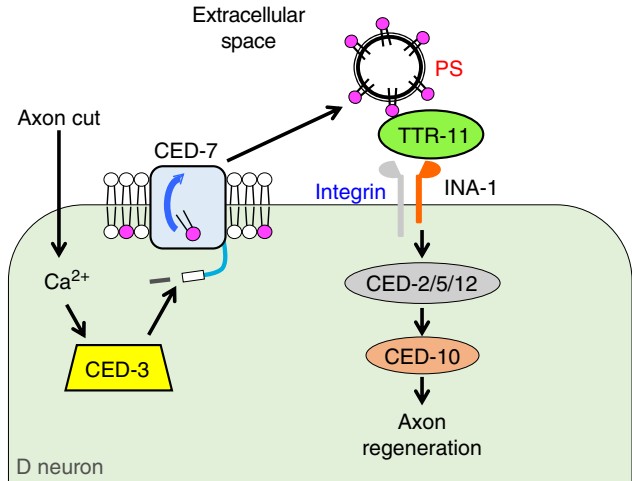

**Fig. 7** Schematic model for the regulation of axon regeneration by caspase-dependent PS exposure. Axonal injury triggers a rise in intracellular calcium, resulting in the activation of CED-3. This, in turn, activates CED-7, leading to an increase in PS exposure. TTR-11 could act as an extracellular PS acceptor and PS-associated TTR-11 may in turn activate integrin. This signal is relayed intracellularly through the CED-2–CED-5–CED-12 module and CED-10 to initiate axon regeneration

7ΔC) (Fig. 6a) from the *unc-25* promoter was able to suppress the *ced-3* defect in axon regeneration, whereas expression of the wild-type CED-7 was not (Fig. 6b and Supplementary Table 1). This suggests that CED-3 acts upstream of CED-7 activation during axonal regeneration. These results support the possibility that CED-3 activates CED-7 by removing its C-terminal region in axon regeneration.

Pinan-Lucarre et al.[31] have recently demonstrated that axon injury causes a transient increase in intracellular calcium in a manner partially dependent on the *C. elegans* calreticulin CRT-1, which in turn activates CED-3. We confirmed that the *crt-1(bz29)* mutant was defective in axon regeneration of D-type motor neurons following axotomy (Fig. 6b and Supplementary Table 1). Furthermore, we found that PS labeling around injured D neurons following axotomy was weaker in *crt-1(bz29)* mutants compared to wild type (Fig. 5c, d and Supplementary Fig. 8). Thus, PS deposition depends on upstream or parallel calcium signaling. These results suggest that CRT-1 affects the intracellular calcium signals required for CED-3 activation. CED-3, in turn, activates CED-7, which exposes PS on the cell surface to promote axon regeneration.

## Discussion

PS functions as an eat-me signal exposed on the surface of apoptotic cells to attract phagocytic receptors, leading to the initiation of their engulfment[32]. Neumann et al.[16] have recently demonstrated that axon injury in the nervous system of *C. elegans* results in PS exposure, which acts as a save-me signal to promote axon re-connection and fusion and thereby re-establish axonal integrity[16]. In this study, we show that surface exposure of PS functions as a critical signal for triggering the initiation of axon regeneration after injury through apoptotic cell clearance molecules in *C. elegans*.

PS displays an asymmetry in its arrangement in the cell membrane, being normally localized to the inner leaflet of the plasma membrane. However, during apoptosis, PS is exposed on the surface of apoptotic cells and serves as a tag for the engulfment of these cells[13]. Maintenance of PS asymmetry is accomplished by integral membrane transporters that specifically flip, flop, or scramble lipids across the bilayer. P4-ATPases function as phospholipid flippases[22]. ABC transporters translocates lipids from the inner to the outer leaflet[25]. In this study, we demonstrate that the *C. elegans* ABC transporter CED-7 promotes the externalization of PS in response to axon injury. Thus, the presentation of the PS signal is a common molecular mechanism functioning in a variety of biological processes.

How is CED-7 activated in response to axon injury? There must be axon regeneration-specific mechanisms that activate CED-7. A recent study has shown that the caspase CED-3 promotes early events in axon regeneration in *C. elegans*[31]. Interestingly, CED-7 has a CED-3 recognition sequence, DQXD, in its C-terminal region. We show that a *ced-7* mutation in which the CED-3 recognition site, Asp-1691, has been replaced with Ala is defective in axon regeneration. In addition, expression of a truncated form of CED-7 lacking its C-terminal region was able to suppress the *ced-3* defect in axonal regeneration. These results raise the possibility that CED-3 removes the C-terminal domain of CED-7, resulting in its activation (Fig. 7).

Caspase-3 induces cell death in many types of cells, and it also acts in differentiation programs in several atypical denucleated cells, such as platelets and lens cells[33]. We would not expect activation of CED-3 to induce apoptotic cell death during the course of axon regeneration in *C. elegans*. Regulation of axon regeneration would require only that CED-3 is activated locally at the injury site. Axonal injury triggers a rapid rise in intracellular

calcium, starting at the injury site[34]. Severed axons may thus trigger activation of the core apoptotic proteins CED-4/Apaf-1 by calcium signaling, mediated by the calreticulin CRT-1 (ref.[31]). The large increase in intracellular calcium amplified by CRT-1 is restricted only to the region where CED-3 is activated. This, in turn, activates CED-7, leading to an increase in PS exposure. Since it is known that caspases play non-apoptotic roles in the nervous system[35], our results suggest that the localized deployment of caspase activities in axonal regeneration may be conserved in higher organisms.

It is likely that laser ablation causes some damage to the surrounding tissue. However, we observed that CED-7 expressed in neurons other than the injured ones is dispensable for axon regeneration. Furthermore, in ced-7 mutants expressing ced-7 in touch neurons, simultaneous laser ablation of axons of both D and touch neurons rescues the regeneration defect of D neurons. These results suggest that CED-7 expressed in the damaged touch neuron induces PS exposure, which acts on the nearby damaged D neuron to induce regeneration. Thus, damage of the tissue surrounding the injured neuron is not necessary to initiate axon regeneration. Since PS signaling and regeneration are not completely abolished in ced-7 mutants, CED-7 may function as one of several components in the regulation of axon regeneration.

In C. elegans, phagocytosis of dying cells is regulated by several conserved intracellular signaling molecules. INA-1/integrin-α recognizes dying cells and transduces a signal to the guanine nucleotide exchange factor complex, CED-2/CrkII, CED-5/DOCK180, and CED-12/ELMO, which activates the GTPase CED-10/Rac, leading to the cytoskeleton reorganization required for engulfment[12]. Remarkably, we have recently shown that this axonal regeneration process involves an apoptotic engulfment machinery that includes these same components[11]. MFG-E8 is a secreted bridging molecule found in mammals that cross-links apoptotic cells to phagocytes by interacting with both PS and the integrin[14]. Importantly, expression of the MFG-E8 C2 domain in D motor neurons inhibits axon regeneration in a manner dependent on its PS-binding activity. On the other hand, a chat-1 mutation that is defective in flippase activity rescues the axon regeneration defect of ced-7 mutants, which otherwise fail to respond to axonal injury due to a defect in cell surface PS exposure. These results provide functional evidence that PS exposure induced by axon injury promotes axon regeneration. The CED-10 GTPase acts in the engulfing cell rather than the dying cell to mediate corpse phagocytosis, which is a non-autonomous mechanism[12]. Externalized PS is produced by the dying cells and triggers a response by the engulfing cells. In contrast, CED-10 regulates axon regeneration in a cell-autonomous manner, as axotomized D-type motor neurons externalize PS, which acts on itself.

How does PS exposure on the axonal cell surface promote axon regeneration? In this study, we identify the ttr-11 gene as critical to this process in C. elegans. TTR-11 is a secreted protein and acts in the INA-1 signaling pathway to promote axon regeneration. Thus, PS-associated TTR-11 may act as a ligand for INA-1 activation. These findings provide evidence that TTR-11 associates with both exposed PS and INA-1, after which this signal is relayed intracellularly through the CED-2–CED-5–CED-12 module and CED-10 (Fig. 7). Interestingly, Mapes et al.[28] have previously demonstrated that CED-7 mediates the release of PS from dying cells during apoptosis. This release of PS by CED-7 requires TTR-52 as an extracellular PS carrier. It is plausible to think that, akin to what occurs during apoptosis, CED-7 could promote PS exposure following axon injury and that TTR-11 could act as an extracellular PS acceptor or carrier to facilitate PS exposure. In support of this possibility, localization of PS around the injured neurons following axon injury is reduced significantly in the ttr-11 mutant. Subsequently, PS-associated TTR-11 may activate integrin and thereby initiate axon regeneration. Thus, this study reveals that components in the apoptotic cascade can also play a novel protective role in axonal regeneration.

## Methods

**Caenorhabditis elegans strains.** The C. elegans strains used in this study are listed in Supplementary Table 2. All strains were maintained on nematode growth medium (NGM) plates and fed with bacteria of the OP50 strain by the standard method[36].

**Plasmids.** To make Pmec-7::ttr-11 and Punc-25::ttr-11, a DNA corresponding to ttr-11 cDNA was synthesized (Eurofins) and then subcloned into pPD52.102 and pSC325, respectively. To clone the Pttr-11::ttr-11 DNA, approximately 2.4 kb of the ttr-11 gene was amplified from the N2 genome by PCR using oligonucleotides 5′-GGATGAAGGGGAACGAGGCTTC-3′ and 5′-GCACGGACTTTGTAGTTGG CTCAG-3′. The Pttr-57::ttr-57 DNA was made by inserting 2.2 kb of the ttr-57 gene, which was amplified from the N2 genome by PCR using oligonucleotides 5′-CCTTCGACCTCCACGAAACTGA-3′ and 5′-CGGCACCGCCGACACCA GTATT-3′, into the pCR2.1 TOPO vector (Invitrogen). The Pttr-11::ttr-11(N46A) was generated by oligonucleotide-directed PCR using Pttr-11::ttr-11 as a template and verified by DNA sequencing. The Pttr-11::nls::venus was made by inserting the ttr-11 promoter, which was amplified with oligonucleotides 5′-ACGGTACCG GAAATGACAGCTG-3′ and 5′- CCGGATCCTTTCTTTGCAAAGATTCG -3′, into the nls::venus vector[18]. Pttr-11::ttr-11::gfp was made by inserting ttr-11 genome DNA amplified with oligonucleotides 5′-ACGGTACCGGAAAATGA-CAGCTG-3′ and 5′-GGTACCGTTTCGTGGAGGTCGAAGGTCAC-3′, into a GFP expression vector pPD95.75. To make Phsp::ss::mfg-e8-c2::gfp, the C2 region of human MFG-E8 was amplified from pMD18-T-MFG-E8 cDNA (Sino Biological) using oligonucleotides 5′-GGTACCAAACGGATGCGCCAATCCCCT-3′ and 5′-GGTACCGAACAGCCTAGCAGCTCCAGG-3′, inserted into the pCR2.1 vector and the DNA sequence was verified. Then the KpnI fragment of the cDNA was inserted into Phsp::ss::gfp, which was generated by inserting the hsp-16.2 promoter derived from the pPD49.78 vector into the pPD95.85 vector. The Phsp::ss::mfg-e8-c2(AAA)::gfp was generated by oligonucleotide-directed PCR using Phsp::ss::mfg-e8-c2::gfp as a template and verified by DNA sequencing. Punc-25::ss::mfg-e8-c2:: gfp was made by replacing the hsp promoter with the unc-25 promoter in the pSC325 vector. To make Punc-25::ced-7 and Pmec-7::ced-7, the ced-7 cDNA was amplified from a pACT C. elegans cDNA library[37], inserted into the pCR2.1 vector, and then inserted into pSC325 and pPD52.102, respectively. Punc-25::ced-7 (D1691A) and Punc-25::ced-7ΔC were generated by oligonucleotide-directed PCR using Punc-25::ced-7 as a template and verified by DNA sequencing. The Pgcy-8:: rfp plasmid was a gift from Dr. Ikue Mori (Nagoya University). The Phsp::ss::anxv:: gfp (pJM31) plasmid was purchased from Addgene. The Punc-25::nes::cfp plasmid has been used to visualize axons[37], while the Pmyo-2::dsred-monomer and Pttx-3:: gfp plasmids have been used as injection markers[38,39]. The TTR-11::FLAG and TTR-11(N46A)::FLAG plasmids for the expression in mammalian cells were made by amplifying a DNA fragment from synthesized ttr-11 cDNA or ttr-11(N46A) cDNA using the oligonucleotides 5′-GGTACCGCTAGCATGAACGCGA-CAATTTTT
CTCGTGG-3′ and 5′-GGTACCGTTTCGTGGAGGTCGAAGGTCAC-3′, and inserted into the pCMV-(DYKDDDDK)-C vector (Clonetech). The INA-1ECD:: GFP plasmid was made by amplifying a DNA fragment from the pACT C. elegans cDNA library[40] using the oligonucleotides 5′-CTCGAGATGCGTGAATGTA TAATTAGCTGGAC-3′ and 5′-GGATCCCCGATAGGTCGAGAGTCTCC AATTGT-3′, and then inserted into the pEGFP-N1 vector (Clonetech).

**Generation of ttr-11 and ttr-57 mutations.** The ttr-11(km64) and ttr-57(km85) deletion mutants were generated using the CRISPR-Cas9 system[41]. The pU6::ttr-11_sgRNA and pU6::ttr-57_sgRNA were made by replacing the unc-119 target sequence of pU6::unc-119_sgRNA (Addgene) with 5′-ACGGGATCCGTACATA TCCG-3′ and 5′-AAACGGATACTTTTCCTTGGAAGG-3′, corresponding to the genomic sequence within the ttr-11 and ttr-57 genes, respectively. The pU6::ttr-11_sgRNA (50 ng μl⁻¹) and pU6::ttr-57_sgRNA (50 ng μl⁻¹) were co-injected together with the Peft-3::cas9-sv40_nls::tbb-2 3′UTR (30 ng μl⁻¹) and Pmyo-2:: dsred-monomer (25 ng μl⁻¹) plasmids into KU501 and KU64 strains, respectively. Each F1 animal carrying a transgene was picked and genomic DNA from its descendants was subjected to a heteroduplex mobility assay[42] to detect the presence of short insertions or deletions in the ttr-11 or ttr-57 genes. The descendants of these animals were selected to obtain the respective homozygous mutant. The ttr-11 (km64) mutation is a 16 bp deletion in the ttr-11 gene, causing a frame shift and premature stop codon in exon 2. The ttr-57(km85) mutation is a 9 bp in-frame deletion in the ttr-57 gene.

**Transgenic animals.** Transgenic animals were obtained by the standard C. elegans microinjection method[43]. Pmyo-2::dsred-monomer, Punc-25::ced-10(G12V), Punc-25::ced-10(T17N), Pttr-11::ttr-11, Pttr-11::ttr-11(N46A), Pmec-7::ttr-11, Punc-25:: ttr-11, Pttr-11::nls::venus, Phsp::ss::mfg-e8-c2::gfp, Phsp::ss::mfg-e8-c2(AAA)::gfp,

*Punc-25::ced-7*, *Punc-25::ced-7(D1691A)*, *Punc-25::ced-7ΔC*, *Pmec-7::ced-7*, *Pttr-11:: ttr-11::gfp*, *Punc-25::ss::mfg-e8-c2::gfp*, *Phsp::ss::anxv::gfp*, and *Pttr-57::ttr-57* plasmids were used in *kmEx466* [*Punc-25::ced-10(G12V)* (25 ng μl⁻¹) + *Pmyo-2::dsred-monomer* (25 ng μl⁻¹)], *kmEx467* [*Punc-25::ced-10(T17N)* (25 ng μl⁻¹) + *Pmyo-2:: dsred-monomer* (25 ng μl⁻¹)], kmEx748 [*Pttr-11::ttr-11* (50 ng μl⁻¹) + *Pmyo-2:: dsred-monomer* (25 ng μl⁻¹)], *kmEx749* [*Pttr-11::ttr-11(N46A)* (50 ng μl⁻¹) + *Pmyo-2::dsred-monomer* (25 ng μl⁻¹)], kmEx750 [*Pmec-7::ttr-11* (50 ng μl⁻¹) + *Pmyo-2::dsred-monomer* (25 ng μl⁻¹)], *kmEx784* [*Punc-25::ttr-11* (50 ng μl⁻¹) + *Pmyo-2::dsred-monomer* (25 ng μl⁻¹)], *kmEx780* [*Pttr-11::nls::venus* (50 ng μl⁻¹) + *Pgcy-8::rfp* (25 ng μl⁻¹)], *kmEx770* [*Phsp::ss::mfg-e8-c2::gfp* (50 ng μl⁻¹) + *Pmyo-2:: dsred-monomer* (25 ng μl⁻¹)], *kmEx771* [*Phsp::ss::mfg-e8-c2(AAA)::gfp* (50 ng μl⁻¹) + *Pmyo-2::dsred-monomer* (25 ng μl⁻¹)], *kmEx761* [*Punc-25::ced-7* (50 ng μl⁻¹) + *Pmyo-2::dsred-monomer* (25 ng μl⁻¹)], *kmEx762* [*Punc-25::ced-7(D1691A)* (50 ng μl⁻¹) + *Pmyo-2::dsred-monomer* (25 ng μl⁻¹)], *kmEx766* [*Punc-25::ced-7ΔC* (50 ng μl⁻¹) + *Pmyo-2::dsred-monomer* (25 ng μl⁻¹)], *kmEx779* [*Pmec-7::ced-7* (50 ng μl⁻¹) + *Pmyo-2::dsred-monomer* (25 ng μl⁻¹)], *kmEx781* [*Pttr-11::ttr-11::gfp* + *Pgcy-8::rfp* (25 ng μl⁻¹)], *kmEx782* [*Punc-25::ss::mfg-e8-c2::gfp* (50 ng μl⁻¹) + *Pmyo-2::dsred-monomer* (25 ng μl⁻¹)], *kmEx783* [*Phsp::ss::anxv::gfp* (50 ng μl⁻¹) + *Pmyo-2::dsred-monomer* (25 ng μl⁻¹)] and *kmEx785* [*Pttr-57::ttr-57* (50 ng μl⁻¹) + *Pmyo-2::dsred-monomer* (25 ng μl⁻¹)], respectively. To make *kmIs10*, *Punc-25::nes:: cfp* (50 ng μl⁻¹) was co-injected with *Pttx-3::gfp* (50 ng μl⁻¹) and then integrated by treatment of animals with 40 Gy from an γ-ray source. The *juIs76* and *wpIs36* integrated arrays were obtained from strains in CGC and outcrossed several times[18,38]. The rescue experiments were done using overexpression constructs. The level of expression may differ for each transgene array.

**Axotomy**. L4 hermaphrodite animals were randomly picked and immobilized with 0.7% sodium azide or 20 mM levamisole solution in M9 buffer on a 2% agarose pad under a cover slip. D-type motor neurons expressing GFP were imaged with a fluorescence microscope. Axons of selected D-type neurons were severed using a 440-nm MicroPoint ablation Laser System from Photonic Instruments. L4 stage animals were randomly picked, subjected to axotomy, transferred to an agar plate for recovery, and then remounted for fluorescent imaging ~24 h after surgery. Axons that grew a distance of 5 μm or more were scored as regenerated axons. Proximal axon segments that showed no change after 24 h were counted as no regenerated axons. For most experiments, photos were taken that make it possible to re-evaluate the data in a blinded manner. To achieve statistical significance, at least 20 living animals having 1–2 axotomized commissures were observed for most experiments. Axotomy was performed in 3–6 trials and 10–20 axons were cut in each trial.

**Heat-shock treatment**. Transgenes driven by the heat-shock promoter were induced by incubating the animals at 37 °C for 30 min, followed by recover at 20 °C for 4 h before continuing with observation or axotomy.

**Microscopy**. Transgenic animals were observed under a ×100 objective of a Nikon ECRIPSE E800 fluorescent microscope and photographed with a Hamamatsu 3CCD camera. Confocal fluorescent images were taken on Olympus FV-500 and a Zeiss LSM-800 confocal laser-scanning microscopes with ×100 and ×63 objectives, respectively.

**Quantification of MFG-E8-C2::GFP**. Intensities of MFG-E8-C2::GFP fluorescence were quantified using the ImageJ program (NIH). First, a square region of interest (ROI) corresponding to an area of $15 \times 15 \, \mu m^2$ and centered on the site of injury was determined for each image and the mean fluorescent intensity of GFP was measured for this ROI to obtain fluorescent intensity per area (FIA). Next, the background was estimated by measuring the mean GFP intensity of an adjacent region with the same $15 \times 15 \, \mu m^2$ size (FIAback). The relative fluorescent intensity per area was obtained by dividing the averaged (FIA-FIAback) obtained from 20 images for each strain with the averaged (FIA-FIAback) from 20 images for the uncut N2.

A complementary approach to quantify the extent of MFG-E8-C2::GFP accumulation around the injury site was taken by estimating the relative area (%) in the ROI with fluorescent intensity above a certain threshold. First, all images were calibrated to have the same mean fluorescent intensity based on the overall mean intensity for the whole stack. Next, for each image, the percentage of area within each ROI that had at least a 200% increase in fluorescent intensity over the mean intensity of the entire image (Analyze > Set Measurements: Area, Limit to Threshold) was measured. All fluorescent images were taken under the same conditions. Experiments were performed in 2–4 trials.

**Immunoprecipitation**. For immunoprecipitation, human HEK293 cells (mycoplasma-free) were transfected with INA-1-ECD::GFP and TTR-11-FLAG (wild type or the N46A mutant), separately, by using FuGENE6 transfection reagent (Promega). After 48 h, cells were lysed in a RIPA buffer containing 50 mM Tris-HCl (pH 7.4), 150 mM NaCl, 0.25% deoxycholic acid, 1% NP-40, 1 mM EDTA, 2 mM dithiothreitol, 1 mM phenylmethylsulfonyl fluoride, phosphatase inhibitor cocktail 2 (Sigma), and protease inhibitor cocktail (Sigma), followed by centrifugation at 15,000 × g for 12 min. Each supernatant was mixed and added to 10 μl (bed volume) of Dynabeads Protein G (Invitrogen) coated with anti-FLAG (M2, mouse monoclonal, Sigma) or anti-GFP (598, rabbit polyclonal, MBL) antibody and rotated for 1 h at 4 °C. The beads were washed three times with ice-cold phosphate-buffered saline and subjected to immunoblotting.

**Immunoblotting**. Extracts were subjected to sodium dodecyl sulfate-polyacrylamide gel electrophoresis and proteins were transferred to a polyvinylidene difluoride membrane (Hybond-P, GE healthcare). Membranes were incubated with the indicated antibodies using the SNAP i.d. system (Millipore). A Sigma anti-FLAG M2 mouse monoclonal antibody or Clontech anti-GFP JL-8 monoclonal antibody was used as the first antibody, and a goat anti-mouse IgG H&L chain-specific peroxidase conjugate (Merck) was used as the second antibody. Immunoreactive bands were detected by a horseradish peroxidase chemiluminescent substrate reagent kit (Novex ECL, Invitrogen) and exposed to FUJI super RX Medical X-ray film (Fujifilm). The uncropped images are shown in Supplementary Figure 12.

**Lipid-binding assay**. To elute TTR-11::FLAG proteins from the immunoprecipitated samples, 40 μg of 0.1 M glycine-HCl (pH 2.8) was added to 20 μg of protein G-sepharose, which bound TTR-11::FLAG proteins via an anti-FLAG antibody, and the supernatant was immediately neutralized with 6 μl of 1 M Tris-HCl (pH 7.5) and further added 50 μl of buffer A (0.1 M Tris-HCl (pH 7.5), 0.3 M NaCl, 20% glycerol, 2 mM dithiothreitol). For the lipid-binding assay, Membrane Lipid Strips™ (Echelon Biosciences) were incubated with TBS-T (Tris-HCl (pH 7.5), 150 mM NaCl, 0.1% Tween-20) + 3% bovine serum albumin (BSA, Sigma-Aldrich) for 1 h at room temperature. Then the eluted wild-type and mutant TTR-11::FLAG proteins were added to TBS-T + 3% BSA and incubated with the preincubated Membrane Lipid Strips™ for 1 h at room temperature. The membranes were washed with TBS-T three times, and then subjected to immunoblotting using an anti-FLAG antibody.

**Statistical analysis**. All statistical analyses were carried out by using QuickCalcs in the GraphPad Software (http://www.graphpad.com/quickcalcs/). Briefly, confidence intervals (95%) were calculated by the modified Wald's method and two-tailed *P* values were calculated using Fisher's exact test. The unpaired *t* test was performed using a *t* test calculator. We also used the G*Power software[44] to calculate the necessary number of observations of axon regeneration.

**Data availability**. All relevant data are available from the authors.

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

## Acknowledgements

We thank Ikue Mori (Nagoya University), *Caenorhabditis* Genetic Center (CGC), National Bio-Resource Project, and *C. elegans* Knockout Consortium for materials. Some strains were provided by the CGC, which is funded by NIH Office of Research Infrastructure Programs (P40 OD10440). This work was supported by grants from the Ministry of Education, Culture and Science of Japan (to K.M.), the Mitsubishi Foundation (to K.M.), Grant-in-Aid for Scientific Research on Innovative Areas (Homeostatic regulation by various types of cell death: 17H05501) from MEXT (to N.H.) and the Project for Elucidating and Controlling Mechanisms of Aging and Longevity from Japan Agency for Medical Research and Development, AMED, under Grant Number JP18gm5010001 (to N.H.). We also thank S. Nagata for helpful suggestions.

## Author contributions

N.H. and K.M. designed the experiments, analyzed data, and wrote the manuscript. N.H., A.T., S.I.P., T.S., and H.H. performed experiments.

## Additional information

**Competing interests:** The authors declare no competing interests.

