## [Peer Review File · Nature Communications]

Reviewers' comments:

Reviewer #1 (Remarks to the Author):

In this paper, Hisamoto and all describe the identification and characterization of ttr-11 as a previously unidentified link between injury and axon regeneration. The authors found that ttr-11 is required cell-non-autonomously to link exposed phosphatidylserine and integrin, thus promoting axon regeneration.

This is a novel and interesting finding. The conclusions support the presented data, the statistical analyses are appropriate and the paper is clearly written. Addressing a number of minor points would add significantly to the thoroughness of this investigation.

- 1) The beginning of the paper mentions that ttr-11 was identified as a suppressor of vhp-11. How vhp-11 interacts with the characterized PS and/or INA-1 pathways would be a valuable addition.
- 2) What is the svh-13 RNAi phenotype? Does loss of ttr-11 function account for all of the regeneration phenotype or does ttr-57 regulate regeneration synthetically with ttr-11?
- 3) The P_{ttr-11::NLS::GFP} expression pattern appears somewhat cytoplasmic. If it is not nuclear, the accuracy of the localization pattern might also be compromised.
- 4) For a thorough analysis of ttr-11 expression and its role in regeneration, the ttr-11 expression pattern before and after injury should be compared.
- 5) ttr-11 also appears to interact with PI4P, PIP2 and PIP3 in a Asn-46 dependent manner (Fig 2b). Is this amount of binding significant? The interactions and their significance should be mentioned in the text for clarity.
- 6) The MFG-E8-C2::GFP images do not immediately appear to reflect the reported quantifications. For example, Line 208 states "no fluorescence was detected in wild-type animals expressing MFG-E8-C2(AAA)::GFP". However, there appears to be a large diffuse area of MFG-E8-C2(AAA)::GFP in cut axons compared to MFG-E8-C2::GFP in uncut axons in figure 4a. The bright area of GFP in the lower left corner of the crt-1 micrograph in figure 5c is also difficult to interpret. To clarify whether localization and/or quantity of PS is regulated by the various genes, the size of the quantified area should be indicated as well as the location of the control exposure in the images and in the methods.
- 7) The arrowheads do not line up with the proximal stump as reported in the figure legend of figure 4a.
- 8) The MFG-E8-C2::GFP reporter provides the opportunity to investigate the dynamics of PS after injury. At what point in the regeneration response is PS exposure necessary and sufficient to activate regeneration?
- 9) ANOH-1 and CED-8 may be acting redundantly. Either the double mutant should be analyzed or the conclusion on line 226 should be amended to allow for this possibility.
- 10) Is ttr-11 expression sufficient to rescue regeneration in a ced-7 mutant background?

Reviewer #2 (Remarks to the Author):

This is a very interesting paper that speaks to mechanisms of neuronal regeneration that are operative consequent to *C. elegans* axotomy and somewhat unexpectedly engage some of the apoptotic death machinery.

The authors identify PS signals after injury, a secreted TTR-11 receptor that may link PS to an integrin receptor INA-1; and suggest that ABC transporter CED-7 might be instrumental in the PS flip. A tie in to a calcium dependent CED-3 pathway is documented, with some data suggesting CED-3 caspase cleavage of the CED-7 inhibitory domain.

This is an interesting story, that in principle would merit publication in *Nature Communications*. The paper, however, seems premature in depth and rigor of establishing its main conclusions, with many questions left. Focusing on generating data that reinforce conclusions is needed:

General—Although listed in Table 1, the n numbers and the number of trials should be indicated for each panel. In general, regeneration levels can vary from trial to trial. Sometimes, >6 genotypes are assayed in one panel of regeneration data, and it is not clear that data are all from the same trial. Wild type regeneration data from a few panels have a very similar value, hopefully, the author did not use the same experiment multiple times as control. Minimally, all this experimental detail has to be made clear.

Figure 1a) 1a is confusing—How is this region designated svh-13? Authors should clarify the background better.

Figure 1c. Total N numbers for each could be indicated in the bars to strengthen this panel.

**A clearer picture might emerge if the authors also tested whether the autonomous rescue can work, using the *unc-25* or *unc-47* promoters. It would be of interest to test TTR-11 expression in the tissue which has direct interaction with D-type neurons--hypodermis. Moreover, the author should also do a TTR-11 rescue experiment in D-type neurons to exclude the cell-autonomous effect.

A factor not considered is the localized damage—we cannot really expect a 100% clean cut of neuron only with a standard laser (femtosecond laser), there is likely some damage from the surrounding tissue; it might be worth giving that fact a mention when clarifying about the location/subcellular location of the mechanism discussed.

Expression pattern/Supplementary Figure 1. The construct contains an NLS but the localization does not appear to be nuclear for the tissues that are indicated as expressing; the transgene array appears to be expressed also in the intestine, but not in intestinal nuclei? Also, the reporter lacks native introns and exons, which might be critical to tissue-specific expression. Minimally, this needs to be mentioned. Overall, the expression data, which are important in the overall story, are not very compelling and could be tightened up with more rigorous experiments. Also, the number of observations that the images are representative of should be indicated.

I think a big question here not addressed is where is TTR-11 expressed after injury—is there

localized induction? A more biologically faithful reporter might address that question.

**Figure 2. Integrin INA-1 interaction with TTR-11 study in COS7 cells is also confusing relative to relevant localization of domains of the interacting proteins. INA-1 extracellular domain is expressed with normally secreted TTR-11::FLAG in cells. Might these two over-expressed proteins INSIDE COS7 cells interacting by artifact? The authors need to make a more convincing argument –in vivo interaction in *C. elegans* best; another complementary in vitro approach needed at least. The bridging is a major point in the model, so this really needs to be experimentally demonstrated without ambiguity.

Figure 3. a) Model would benefit for indication of what cell this is happening in-neuron (vs. surrounding tissue). b) authors should indicated n number for each bar in panel b.

Figure 4. Interesting study to detect PS in vivo. Authors should indicate whether this particular pattern is representative of how many of the 20 observations in legend. Pattern of distribution seems to be limited to the cut on the right side, and not necessarily at the neuron—why would that be? Is this asymmetry seen in all examples? Is there always signal one hour after cutting? For how long injury is the signal detectable? Ideally these experiments could be repeated with the TTR-11 version of the PS-binding reporter, but the approach is convincing as a means by which to visualize PS.

The MFG-E8-C2::GFP signal is a bit overexposed. It might be better to use “area” other than “relative fluorescent intensity” to quantify the GFP signal.

Panel c. Should include N number in legend or on bars.

Figure 5 studies. **Issue of scramblases *anoh-1* and *ced-8*: an obvious possibility is that these might be redundant for scramblase activity. The authors should repeat the experiment in a double mutant to really rule out these are involved. This is particularly important because the exposure of PS is a core finding of the paper—if the double changes the outcome, the overall conclusion would be further solidified and mechanistic information would be gleaned.

Similarly, although the authors show that the *chat-1* mutant suppresses *ced-7* mutants, it would strengthen the conclusions if the *chat-1* was shown to increase the PS signal detection in this model, after axotomy.

What does a *chat-1* mutation do to the baseline regeneration phenotype?

The question of cell autonomy for *ced-7* could be better addressed, since in apoptosis *ced-7* has been said to function in both the dead and engulfing cells. *ced-7* might be expressed in all cells but neurons (for example *sid-1* promoter) to test if needed outside the injured neuron. Rescue in each candidate involved tissue should be accomplished.

Figure 6. add n numbers to bar.

Figure 7 and Discussion. I think the authors might better address in the study, and in the

discussion, the sites of action of the molecules they identified. If MFG-E8 normally bridges dying and engulfing cells together by linking PS (dying cells) with integrin receptor (engulfing cell); in regeneration, the data support that the neuron makes PS and that TTR-11 is supplied externally.

Figure 7 suggests different responses at the distal and proximal ends of the cut. It is not clear if this is reasonable. An odd observation in the field is that initially both truncated ends can initiate a growth response. The authors might be encouraged to think about whether the initial signaling takes place across the break—similarly on both sides.

Other points to consider, though not essential:

Transgenic lines studied are all extrachromosomal arrays –high copy number and possibly unstable in that copy number. This is a general problem—ambiguities could be addressed by low copy number MOSCII approaches.

The question as to whether caspase cleavage occurs for CED-7 in vivo has not been biochemically addressed; model would be more compelling with in vivo support of the actual cleavage event.

Figure 2. The N46A point mutation is expected to reduce TTR-11 binding to PS and PA. The author presented the binding data of wild type vs mutant on two different blots, but there is strong difference on the background of the two blots. The authors can show the difference on same blot and have some statistical assay, then the conclusion would be much more solid. There could be some other factors that may increase the variation of data presentation, such as the expression level of WT TTR-11 vs N46A mutant. The author might adjust the sample volume to be used in this test according to the TTR-11 protein level, which may be measured by FLAG IB.

2c. Is the same WCE used for both FLAG IP and IB of WCE? What are the repeat trial numbers?

In general, the double mutant strategy is used to suggest action in a common pathway. This argument is best for null mutation doubles. One issue is there could be a “ceiling” and “basement” effect—things could not go much lower than the singles. The authors might want to mention the assumptions that hold for the double genetic mutants.

Writing.

Line 42. At the authors note in the second sentence, peripheral neurons regeneration, but CNS neurons do not; there fore first sentence is misleading. Better: A fundamental and conserved property of some neurons....

Line 55 C. elegans' rather than Its

Line 64 stimuli, generating

Line 65 These

Line 525 and the DNA sequence was verified

Line 139 the ttr-11 defect in D motor neurons

Line 140 is most likely required for axonal regeneration non-autonomously.

Line 358 cells and is recognized

Supplementary Table 2 KU723 should not have a question mark

Probably do not need periods at the end of each strain name

Point-by-point response to the referees' comments

Paper: NCOMMS-17-19090-T

Authors: Hisamoto et al.,

Title: Phosphatidylserine exposure mediated by caspase activation of ABC transporter activates the integrin signaling pathway promoting axon regeneration

The manuscript has been revised in accordance with the comments raised by three referees. Responses to the comments are as follows:

Our responses to the comments of Reviewer #1

1) The beginning of the paper mentions that *ttr-11* was identified as a suppressor of *vhp-11*. How *vhp-11* interacts with the characterized PS and/or INA-1 pathways would be a valuable addition.

C. elegans axon regeneration is regulated by the JNK MAPK pathway. The JNK cascade can be inactivated by the MAPK phosphatase VHP-1 and a *vhp-1* loss-of-function mutation causes hyperactivation of the JNK pathway. In *vhp-1* null mutants, development is arrested at the early larval stage, and this phenotype is suppressed by mutations in *mlk-1*, *mek-1* or *kgb-1*. To identify additional components involved in JNK-mediated signaling, we undertook a genome-wide RNAi screen for suppressors of *vhp-1* lethality. It can be expected that survival would be due to RNAi downregulation of a gene that contributes to JNK pathway activity. We have now described this point in the text (p. 6) with Supplementary Figure 1.

2) What is the *svh-13* RNAi phenotype? Does loss of *ttr-11* function account for all of the regeneration phenotype or does *ttr-57* regulate regeneration synthetically with *ttr-11*?

The *svh-13* RNAi showed no obviously altered phenotype with respect to growth, movement or egg laying.

The Asp-51 residue in TTR-52 is essential for binding to PS and the function in apoptotic cell engulfment. The corresponding site (Asn-46) is conserved in TTR-11 and plays a crucial role in axon regeneration. In contrast, TTR-57 does not contain this conserved site (Supplementary Figure 4). Therefore, it is unlikely that TTR-57 regulates regeneration synthetically with TTR-11.

Since the *ttr-11* and *ttr-57* genes are located next to one another, it is very difficult to construct the double mutation.

3) The *Pttr-11::NLS::GFP* expression pattern appears somewhat cytoplasmic. If it is not nuclear, the accuracy of the localization pattern might also be compromised.

We have reexamined the *ttr-11* expression pattern. We show that a *Pttr-11::nls::gfp* reporter is expressed in several tail neurons (Supplementary Figure 3). We have rewritten this point in the text (p. 7) and replaced the data

in Supplementary Figure 3.

We have tried to examine the localization pattern of TTR-11 proteins using a protein fusion gene. However, fusion of GFP to either the C-terminus or the N-terminus after the signal sequence of *ttr-11* did not produce a functional fusion gene.

Therefore, we asked whether the *ttr-11* mutation would affect PS accumulation around the axon segments of D-type motor neurons following axotomy. We show that localization of MFG-E8-C2::GFP around the injured neurons following axon injury was lower in the *ttr-11* mutant versus wild-type (Figure 5c, d and Supplementary Figure 5). These results suggest that TTR-11 may localize around injured neurons and that PS accumulation is dependent on the presence of TTR-11.

4) For a thorough analysis of *ttr-11* expression and its role in regeneration, the *ttr-11* expression pattern before and after injury should be compared.

The *ttr-11* expression pattern was not affected by axon injury. We now mention this point in the text (p. 7).

5) *ttr-11* also appears to interact with PI4P, PIP2 and PIP3 in a Asn-46 dependent manner (Fig 2b). Is this amount of binding significant? The interactions and their significance should be mentioned in the text for clarity.

We show that expression of MFG-E8-C2::GFP inhibited axon regeneration. By contrast, expression of MFG-E8-C2(AAA)::GFP, a variant that does not bind PS, did not appear to affect regeneration (Figure 4c). These results are consistent with a role for exposed PS in axon regeneration. Thus, although TTR-11 binds to PI4P, PI(3,4)P₂ and PI(3,4,5)P₃, the interaction of TTR-11 with PS is likely important for axon regeneration. We have mentioned this point in the text (p. 10~11).

6) The MFG-E8-C2::GFP images do not immediately appear to reflect the reported quantifications. For example, Line 208 states no fluorescence was detected in wild-type animals expressing MFG-E8-C2(AAA)::GFP. However, there appears to be a large diffuse area of MFG-E8-C2(AAA)::GFP in cut axons compared to MFG-E8-C2::GFP in uncut axons in figure 4a. The bright area of GFP in the lower left corner of the crt-1 micrograph in figure 5c is also difficult to interpret. To clarify whether localization and/or quantity of PS is regulated by the various genes, the size of the quantified area should be indicated as well as the location of the control exposure in the images and in the methods.

As suggested, we have taken a complementary approach to quantify the extent of MFG-E8-C2::GFP accumulation around the injury site by estimating the relative area in the ROI with fluorescent intensity above a certain threshold. We have added these data in Supplementary Figure 5. We have described this quantification in “Methods”.

We have rewritten our description of the localization pattern of MFG-E8-C2(AAA)::GFP (p. 10).

The bright area of GFP in the lower left corner of the *crt-1* micrograph in Figure 5c reflects autofluorescence of the intestine.

7) The arrowheads do not line up with the proximal stump as reported in the figure legend of figure 4a.

The arrowheads indicate the sites of laser surgery. We have rewritten the figure legend of Figure 4a.

8) The MFG-E8-C2::GFP reporter provides the opportunity to investigate the dynamics of PS after injury. At what point in the regeneration response is PS exposure necessary and sufficient to activate regeneration?

We have monitored the dynamics of MFG-E8-C2::GFP localization after D-type neuron axotomy. We show that at 10 min after axotomy, MFG-E8-C2::GFP appears between the proximal and distal axon segments and that this localization of MFG-E8-C2::GFP began diffusing by 2 hr after surgery (Supplementary Figure 6).

9) ANOH-1 and CED-8 may be acting redundantly. Either the double mutant should be analyzed or the conclusion should be amended to allow for this possibility.

As suggested, we have constructed *anoh-1; ced-8* double mutants. We show that the double mutation had no effect on axon regeneration (Figure 5b and Supplementary Table 1). This result excludes the possibility that ANOH-1 and CED-8 act redundantly.

10) Is *ttr-11* expression sufficient to rescue regeneration in a *ced-7* mutant background?

As suggested, we have tested the effect of *ttr-11* overexpression on the *ced-7* defect in axon regeneration. We show that overexpression of the *ttr-11* gene was able to suppress the *ced-7* mutation (Supplementary Figure 7 and Supplementary Table 1).

Our responses to the comments of Reviewer #2

General; Although listed in Table 1, the n numbers and the number of trials should be indicated for each panel. In general, regeneration levels can vary from trial to trial. Sometimes, >6 genotypes are assayed in one panel of regeneration data, and it is not clear that data are all from the same trial. Wild type regeneration data from a few panels have a very similar value, hopefully, the author did not use the same experiment multiple times as control. Minimally, all this experimental detail has to be made clear.

As suggested, we have indicated the numbers of axons examined in each panel of figure.

Under our assay conditions for axon regeneration, when the same person performs the assay, regeneration levels do not vary from trial to trial. However, when different people perform the assay for the same sample, regeneration levels vary slightly from person to person.

Axotomy was performed in 3~6 trials and 10~20 axons were cut per trial. We now describe this experimental detail in “Methods” (p. 26).

Figure 1a; 1a is confusing. How is this region designated svh-13? Authors should clarify the background better.

To identify additional components involved in JNK-mediated signaling, we undertook a genome-wide RNAi screen for suppressors of *vhp-1* lethality. The RNAi clones discovered by this screen were termed *svh*, for suppressors of *vhp-1* lethality. The *svh-13* RNAi clone corresponds to the F46B3.3 & F46B3.18 RNAi clone. We now clarify this point in the text (p. 6).

Figure 1c; Total N numbers for each could be indicated in the bars to strengthen this panel.

As suggested, we have indicated the total N numbers for each bar in Figure 1c panel.

****A clearer picture might emerge if the authors also tested whether the autonomous rescue can work, using the *unc-25* or *unc-47* promoters. It would be of interest to test TTR-11 expression in the tissue which has direct interaction with D-type neurons-hypodermis. Moreover, the author should also do a TTR-11 rescue experiment in D-type neurons to exclude the cell-autonomous effect.**

As suggested, we have tested whether autonomous rescue can work, using the *unc-25* promoter. We show that the *ttr-11* defect in axon regeneration of D motor neurons was not rescued by expression of *ttr-11* in D-type motor neurons by the *unc-25* promoter (Supplementary Figure 2 and Supplementary Table 1).

A factor not considered is the localized damage; We cannot really expect a 100% clean cut of neuron only with a standard laser (femtosecond laser), there is likely some damage from the surrounding tissue. It might be worth giving that fact a mention when clarifying about the location/subcellular location of the mechanism discussed.

We now discuss this point in the “Discussion” section (p. 17~18).

Expression pattern/Supplementary Figure 1; The construct contains an NLS but the localization does not appear to be nuclear for the tissues that are indicated as expressing; the transgene array appears to be expressed also in

the intestine, but not in intestinal nuclei? Also, the reporter lacks native introns and exons, which might be critical to tissue-specific expression. Minimally, this needs to be mentioned. Overall, the expression data, which are important in the overall story, are not very compelling and could be tightened up with more rigorous experiments. Also, the number of observations that the images are representative of should be indicated.

We have reexamined the *ttr-11* expression pattern. We show that a *Pttr-11::nls::gfp* reporter is expressed in several tail neurons (Supplementary Figure 3). We have rewritten this point in the text (p. 7) and replaced the data in Supplementary Figure 3.

It is likely that the fluorescent signal observed in the intestine reflects autofluorescence.

As suggested, we now mention that a reporter containing native introns and exons might be critical to tissue-specific expression (p. 7).

As suggested, we have indicated the number of observations in the legend of Supplementary Figure 3.

I think a big question here not addressed is where is TTR-11 expressed after injury. Is there localized induction? A more biologically faithful reporter might address that question.

The *ttr-11* expression pattern was not affected by axon injury. We now mention this point in the text (p. 7).

We have tried to examine the localization pattern of TTR-11 proteins using a protein fusion gene. However, fusion of GFP to either the C-terminus or the N-terminus after the signal sequence of *ttr-11* did not produce a functional fusion gene.

Therefore, we asked whether the *ttr-11* mutation would affect PS accumulation around the axon segments of D-type motor neurons following axotomy. We show that localization of MFG-E8-C2::GFP around the injured neurons following axon injury was lower in the *ttr-11* mutant versus wild-type (Figure 5c, d and Supplementary Figure 5). These results suggest that TTR-11 may localize around injured neurons and that PS accumulation is dependent on the presence of TTR-11.

****Figure 2; Integrin INA-1 interaction with TTR-11 study in COS7 cells is also confusing relative to relevant localization of domains of the interacting proteins. INA-1 extracellular domain is expressed with normally secreted TTR-11::FLAG in cells. Might these two over-expressed proteins INSIDE COS7 cells interacting by artifact? The authors need to make a more convincing argument. In vivo interaction in *C. elegans* best. Another complementary in vitro approach needed at least. The bridging is a major point in the model, so this really needs to be experimentally demonstrated without ambiguity.**

As suggested, we have confirmed the interaction between INA-1-ECD::GFP and TTR-11::FLAG in vitro. We expressed INA-1-ECD::GFP and TTR-11::FLAG in HEK293 cells separately. Cell lysates prepared from each cell were mixed in vitro and immunoprecipitated with FLAG antibody. We show that INA-1-ECD::GFP was co-precipitated with TTR-11::FLAG (Figure 2c).

Interestingly, we showed that when we substituted the TTR-11(N46A)::FLAG mutated form for the wild-type, in vitro association with INA-1-ECD::GFP was significantly weaker (Figure 2c). Mammalian MFG-E8 binds to integrin efficiently in the presence of PS. These results suggest that PS binding promotes the association of MFG-E8 and TTR-11 with integrin. These findings provide strong evidence that TTR-11 acts as a bridging molecule that cross-links PS with the INA-1 receptor. We discuss this point in the “Discussion” section (p. 19).

Figure 3; a) Model would benefit for indication of what cell this is happening in-neuron (vs. surrounding tissue). b) authors should indicated n number for each bar in panel b.

As suggested, we have indicated that the cell type used in Figure 3a is the D-type motor neuron.

As suggested, we have indicated the total N numbers for each bar in Figure 3b panel.

Figure 4; Interesting study to detect PS in vivo. Authors should indicate whether this particular pattern is representative of how many of the 20 observations in legend. Pattern of distribution seems to be limited to the cut on the right side, and not necessarily at the neuron. Why would that be? Is this asymmetry seen in all examples? Is there always signal one hour after cutting? For how long injury is the signal detectable? Ideally these experiments could be repeated with the TTR-11 version of the PS-binding reporter, but the approach is convincing as a means by which to visualize PS.

We observed this particular expression pattern in all of our observations. As suggested, we now mention this point in the legend to Figure 4a.

We have monitored the dynamics of MFG-E8-C2::GFP localization after D-type neuron axotomy. We show that at 10 min after axotomy, MFG-E8-C2::GFP expression appeared in a region between the proximal and distal axon segments and that this localization of MFG-E8-C2::GFP began diffusing by 2 hr after surgery (Supplementary Figure 6).

The asymmetrical distribution of MFG-E8-C2::GFP was not observed in all examples. At 10 min after axotomy, the pattern of MFG-E8-C2::GFP distribution is symmetrical (Supplementary Figure 6).

We have tried to examine the localization pattern of TTR-11 proteins using a

protein fusion gene. However, fusion of GFP to either the C-terminus or the N-terminus after the signal sequence of *ttr-11* did not produce a functional fusion gene.

Therefore, we asked whether the *ttr-11* mutation would affect PS accumulation around the axon segments of D-type motor neurons following axotomy. We show that localization of MFG-E8-C2::GFP around the injured neurons following axon injury was lower in the *ttr-11* mutant versus wild-type (Figure 5c, d and Supplementary Figure 5). These results suggest that TTR-11 may localize around injured neurons and that PS accumulation is dependent on the presence of TTR-11.

The MFG-E8-C2::GFP signal is a bit overexposed. It might be better to use other than relative fluorescent intensity to quantify the GFP signal.

As suggested, we have taken a complementary approach to quantify the extent of MFG-E8-C2::GFP accumulation around the injury site by estimating the relative area in the ROI with fluorescent intensity above a certain threshold. We have added these data in Supplementary Figure 5. We have described this quantification in “Methods”.

Panel c; Should include N number in legend or on bars.

As suggested, we have indicated the total N numbers for each bar in Figure 4c panel.

Figure 5 studies; **Issue of scramblases *anoh-1* and *ced-8*. An obvious possibility is that these might be redundant for scramblase activity. The authors should repeat the experiment in a double mutant to really rule out these are involved. This is particularly important because the exposure of PS is a core finding of the paper if the double changes the outcome, the overall conclusion would be further solidified and mechanistic information would be gleaned.

As suggested, we have constructed *anoh-1; ced-8* double mutants. We show that the double mutation had no effect on axon regeneration (Figure 5b and Supplementary Table 1). This result excludes the possibility that ANOH-1 and CED-8 act redundantly.

Similarly, although the authors show that the *chat-1* mutant suppresses *ced-7* mutants, it would strengthen the conclusions if the *chat-1* was shown to increase the PS signal detection in this model, after axotomy.

What does a *chat-1* mutation do to the baseline regeneration phenotype?

As suggested, we have tried to examine whether the *chat-1* mutation would suppress the PS signal in *ced-7* mutants. However, we failed to construct *ced7; chat-1* double mutants carrying *Phsp::ss::mfg-e8-c2::gfp*. The *chat-1* mutation itself does not have significant effect on the frequency of axon regeneration, as shown in Figure 5b, suggesting that *chat-1* does not affect the baseline regeneration phenotype.

The question of cell autonomy for *ced-7* could be better addressed, since in apoptosis *ced-7* has been said to function in both the dead and engulfing cells. *ced-7* might be expressed in all cells but neurons (for example *sid-1* promoter) to test if needed outside the injured neuron. Rescue in each candidate involved tissue should be accomplished.

As suggested, we have examined whether *ced-7* expression is involved in axon regeneration if expressed in cells outside of the injured neurons. To do this, we used the *mec-7* promoter to express the *ced-7* gene in touch neurons. Touch neuron axons run parallel to the body axis and cross almost perpendicular to motor neuron axons (Supplementary Figure 8a). We show that expression of *ced-7* in touch neurons did not rescue the *ced-7* defect in D-type motor neuron regeneration (Supplementary Figure 8b and Supplementary Table 1). This result is consistent with the possibility that CED-7 functions cell-autonomously. On the other hand, when both touch and D neurons were injured simultaneously in *ced-7* mutants expressing *ced-7* in touch neurons, the regeneration defect of D neurons was suppressed (Supplementary Figure 8b and Supplementary Table 1). These results suggest that CED-7 in the damaged touch neuron induces PS exposure, which can also act on the damaged D neuron to regenerate. We now describe these results in the text (p. 12~13).

Figure 6; add n numbers to bar.

As suggested, we have indicated the total N numbers for each bar in Figure 6b panel.

Figure 7 and Discussion; I think the authors might better address in the study, and in the discussion, the sites of action of the molecules they identified. If MFG-E8 normally bridges dying and engulfing cells together by linking PS (dying cells) with integrin receptor (engulfing cell), in regeneration, the data support that the neuron makes PS and that TTR-11 is supplied externally.

We show that localization of MFG-E8-C2::GFP around the injured neurons following axon injury was lower in the *ttr-11* mutant versus wild-type (Figure 5c, d and Supplementary Figure 5). These results suggest that TTR-11 may localize around injured neurons and that PS accumulation is dependent on the presence of TTR-11.

We discuss this point raised by the reviewer in the “Discussion” (p. 19).

Figure 7 suggests different responses at the distal and proximal ends of the cut. It is not clear if this is reasonable. An odd observation in the field is that initially both truncated ends can initiate a growth response. The authors might be encouraged to think about whether the initial signaling takes place across the break-similarly on both sides.

We discuss this point raised by the reviewer in the “Discussion” (p. 19).

Other points to consider, though not essential;

Transgenic lines studied are all extrachromosomal arrays; high copy number and possibly unstable in that copy number. This is a general problem. Ambiguities could be addressed by low copy number MOSCII approaches.

We agree with this suggestion. However, we have not yet succeeded in establishing the MOSCII system in our laboratory.

The question as to whether caspase cleavage occurs for CED-7 in vivo has not been biochemically addressed. Model would be more compelling with in vivo support of the actual cleavage event.

Although this would be informative, it is technically difficult to biochemically examine whether axon injury induces CED-3-dependent cleavage for CED-7 in D-type motor neurons.

Figure 2; The N46A point mutation is expected to reduce TTR-11 binding to PS and PA. The author presented the binding data of wild type vs mutant on two different blots, but there is strong difference on the background of the two blots. The authors can show the difference on same blot and have some statistical assay, then the conclusion would be much more solid. There could be some other factors that may increase the variation of data presentation, such as the expression level of WT TTR-11 vs N46A mutant. The author might adjust the sample volume to be used in this test according to the TTR-11 protein level, which may be measured by FLAG IB.

As pointed out, it is possible that difference in the expression level of wild-type TTR-11 vs TTR-11(N46A) mutant might influence the variation in their PS-binding activities as detected in our assay. For a more statistical assay, we tried to purify GST-fused TTR-11 proteins from *E. coli*, but failed to do so due to protein insolubility. Therefore, we have removed the PS-binding data for TTR-11(N46A) (Figure 2b).

Instead of the PS-binding data for TTR-11(N46A), we have described the effect of TTR-11(N46A) on its in vitro interaction with integrin (Figure 2c). We show that the in vitro association between the TTR-11(N46A)::FLAG mutated form and INA-1-ECD::GFP is significantly lower compared to wild-type TTR-11 (Figure 2c). Mammalian MFG-E8 binds to integrin efficiently in the presence of PS. These results suggest that PS binding promotes the association of MFG-E8 and TTR-11 with integrin.

2c; Is the same WCE used for both FLAG IP and IB of WCE? What are the repeat trial numbers?

We have replaced the in vivo interaction with the in vitro interaction (Figure 2c).

In general, the double mutant strategy is used to suggest action in a common pathway. This argument is best for null mutation doubles. The authors might

want to mention the assumptions that hold for the double genetic mutants.

We agree with this comment. However, we have also presented data showing the epistatic relationship, which should confirm function in a common pathway.

Writing;

Line 42; At the authors note in the second sentence, peripheral neurons regeneration, but CNS neurons do not. Therefore, first sentence is misleading. Better; A fundamental and conserved property of some neurons.

Line 55; C. elegans rather than Its

Line 64; stimuli, generating

Line 65; These

Line 525; and the DNA sequence was verified

Line 139; the ttr-11 defect in D motor neurons

Line 140; is most likely required for axonal regeneration non-autonomously.

Line 358; cells and is recognized

Supplementary Table 2; KU723 should not have a question mark. Probably do not need periods at the end of each strain name.

As suggested, we have made these corrections. We appreciate the reviewer's careful reading.

Reviewers' comments:

Reviewer #1 (Remarks to the Author):

While the points raised in the previous review were largely addressed, there are a few minor points that remain. These should be addressed before publication.

1) *ttr-11* expression pattern: 1) Was the revised expression pattern visualized in the same strain that was analyzed previously? If not, why does this construct chosen as the most accurate representation of the expression pattern? Whether VENUS is still expressed in the cytoplasm of the pharyngeal and vulva cells should be indicated in the text. 2) The weak neuronal expression of *ttr-11* could be masked by intestinal auto fluorescence. Use of an RFP variant would circumvent this problem. Otherwise, since an extrachromosomal construct was used, more than two animals should be analyzed to compensate for mosaic expression. 3) Are the GFP-positive tail neurons a subset of the GABA neurons? 4) It is not clear why reduced MFG-E8-C2::GFP expression in *ttr-11* mutants suggests 'that *ttr-11* may localize around injured neurons'. One might expect that TTR-11 would compete with MFG-E8-C2::GFP if they both bind to any PS that is present. If so, MGF-E8-C2::GFP levels would increase in *ttr-11* mutants. This possibility should be addressed.

ttr-11 rescue: 1) The Punc-25 experiments were done in young adults. Not using the same experimental paradigm as the rest of the study raises the question of why they were done that way. Was regeneration significantly different in young adults and L4 animals in any of the genotypes? 2) The data that *ttr-11* does not function cell autonomously (at least in part) are fairly weak and should be presented as such. For example, there is no significant difference in regeneration between wild type and *ttr-11*; Punc-25::*ttr-11* animals, suggesting *ttr-11* does function in GABA neurons.

ttr-57: RNAi of one (or both) mutant(s) in the background of the other deletion would address the question, and if negative, would strengthen the argument that binding to PS at Asn-46 is necessary for regeneration. Nonetheless, the text only states that the deletion has no effect on regeneration, so is accurate as written. At the least, the presence of *ttr-57* in this figure should be addressed in the text.

Discussion of *ced-7* rescue (line 430): it is possible that *ced-7* expression from neighboring GABA cells rescue regeneration in cut GABA neurons. Revise statement 'CED-7 expressed in other than injured neuron is dispensable for axon regeneration'.

Reviewer #2 (Remarks to the Author):

The authors have reasonably addressed most concerns raised in initial review and this has improved the paper. This paper provides interesting data on the role of phosphatidylserine

and its interacting molecules in pathways that activate regeneration in *C. elegans* motoneurons. I strongly recommend eventual publication in Nature Communications, although in my opinion there are additional science points that need to be addressed prior to acceptance. Most suggestions below are changes that I feel would enhance clarity or rigor and the authors can consider if these changes would accomplish this. Critically important to science issues are marked by asterisks.

Figure 1

b. The authors could add representative of all, most or whatever, for clarity. To my experience this defect shown looks like best case scenario. The authors data in Figure 3 suggest that at best ~75% are scored as regenerating..so they need to be clear on that.

c. The legend should indicate D type neurons because of the odd result with of rescue using the touch neuron specific *mec-7* promoter that is included—one might confuse with touch neuron axotomies.

Figure 2

b. I wonder if the journal will request definition of abbreviations on panel.

Data/story could be enhanced by using interaction-defective TTR-11 mutant.

c. Useful to define N46A as predicted interaction defective. Write N46A on panel.

One wonders about the "reverse" interaction test—IP INA first, then test TTR-11 FLAG

Figure 3

Line 804 is diagram (singular)

Figure 4

a. Legend should indicate that mCherry is used to visualize the neuron

b. Legend should indicate how long after axotomy scores were taken. Clarify the relative of what score to what score.

c. panel is a bit hard to decipher at first glance. Possibly more clear if WT was replaced by MFG-E8-C2::GFP—don't the two WTs indicate different things?

Would have been nice to repeat independently with Annexin V::GFP.

**The biggest issue here is what is happening with the heat shock promoter—do the authors induce with hs, do they rely on the laser?

Figure 5.

c. Do the authors specify uniform exposure for the collection of data in methods. How many are tested, how many is this representative of?

d. N numbers need to be added to the figure panel to indicate the rigor of the data. What is relative to what should be articulated in the legend

Figure 7

Needs much more explanation in the legend.

**I think the model presentation has to be remade as it is misleading. There are no data that generally support that CED-3/CED-7 act in the distal fragment; most commonly, this most likely occurs in the injured neuron (though maybe in surrounding tissue). The authors could state that the current model is for the touch neuron potentiation of DA MN regeneration; but better to diagram the general model, with CED-3/CED-7 events on both sides, or just summarized on the side that regenerates.

Supplemental figures:

S Figure 2. Title is misleading and should be changed. Data support expression in D Mneuron or touch neurons can complement regeneration defects; they do not on their own establish a non-autonomous mechanism. They show that the mechanism CAN be non-autonomous, but it can be autonomous.

**S Figure 3. The expression study seems somewhat incomplete. This is a transcriptional fusion, regulatory and coding sequences may be missing. Can there be a repeat with the intact genomic version of the gene, even if functionality is not supported—is the pattern different? **Looking at two animals is insufficient; what age is this animal, does the pattern change with age?

**There are no data presented here regarding how the reporter responds to injury—information claimed on line 145. The science basis for this statement should be included.

S Figure 5. Indicate increase as compared to what.

S Figure 6. Add how many animals were examined to show a similar time course. Surely this is not a single experiment.

S Figure 8. Can the authors indicate which D neurons are cut in these studies? Are they always the ones closest to the touch neuron? Do the more distant ones regenerate less efficiently than the ones close to the injured PLM?

S Table 2. Some of the nomenclature designations are unclear. Each Ex transgene should be spelled out for genotype in this list.

Text

Line 84. Authors might consider better phrasing: ...does not have a MFG-E8 homolog easily detected by sequence homology.. my point is that there can be functional homologs, which is part of the point of this paper.

Line 101. The authors might be just a bit more careful in stating the conclusion—there is reasonable evidence for this model; but the precise signaling relationship is not proven and

could involve other players, not necessarily in direct interaction. How about: "Our data support a model in which TTR-11....."

Line 121 the figure 1a referenced does not show the reduced expression, so the statement is misleading. Is the expression data in the paper? It could be—refer to it, as supplementary. But I think the intention can be met by changing the statement: svh-13 RNAi is expect to target both ttr-11 and ttr-57 genes.

**Around line 141. As noted above, the expression data as documented are not compelling. The authors site outcome based on two animals, which is not enough. The paper might be enhanced by adding on a full length clone expression and localization, even in the absence of demonstrated functionality. The authors have this construct in hand. One wonders whether testing in single copy might result in functionality, given concerns about potential dominant negative effects when a certain level of overexpression is attained.

**The authors state no change in this construct with injury—where is that data, what are the N numbers there? The legend for S Figure 3 does not show this. A key interesting question is where the TTR-11 moves upon injury—great if that could be addressed.

Another thing that the authors should consider here is the question of gene dosage in these rescues. Virtually all their studies are over-expression construct, and the level of expression may differ for each transgene array. The MFG-E8-C2 can have a dominant negative effect, presumably by binding PS or obstructing access to other interactors..wouldn't the same be potentially expected for elevated TTR-11? Only single lines are reported. The authors should include some mention of this caveat for interpretation of rescue studies in some legend or supplementary site.

Line 154. Better to state that TTR-11 can act non-autonomously for axonal regeneration.

Line 188. Better to state that TTR-11 can act as..

**Page 10, experiments on mfg-e8-c2::gfp. These studies are really cool. Still, the use of the heat shock promoter introduces some potential complications—was heat shock used to induce expression? What is the impact of heat shock on the regeneration response? There is no mention of how the study was conducted with regard to any potential induction, which has to be made clear.

Line 370. Need to restate this sentence as really the data shown regard calcium change/regulation in regard to PS deposition. Summarize that PS depends on upstream or parallel ca signal; then in the following sentence state the whole model.

Discussion

--need to articulate somewhere that this is only one part of the regeneration story—all PS signal is not lost in ced-7 or other backgrounds, all regeneration is not lost either, authors should explicitly state this.

line 464 use secreted rather than secretory

line 467; not clear why TTR-11 N46A should reduce PS binding without more data. PS should also bind TTR-11. Could the TTR-11 be tested as was done for INA-1 for PS binding in the biochemical assay, and use N46A as a control?

Paper: NCOMMS-17-19090A
Authors: Hisamoto et al.,
Title: Phosphatidylserine exposure mediated by caspase activation of ABC transporter activates the integrin signaling pathway promoting axon regeneration

The manuscript has been revised in accordance with the additional comments raised by two referees. Responses to the comments are as follows:

Our responses to the comments of Reviewer #1

ttr-11 expression pattern:

1) Was the revised expression pattern visualized in the same strain that was analyzed previously? If not, why does this construct chosen as the most accurate representation of the expression pattern? Whether VENUS is still expressed in the cytoplasm of the pharyngeal and vulva cells should be indicated in the text.

We further verified the expression pattern of *ttr-11* in animals carrying *Pttr-11::nls::venus* and *Punc-25::nes::cfp*. CFP under the control of the *unc-25* promoter was used to visualize D neurons. We show that VENUS expression was observed in HSN neurons, excretory gland cells, hypodermal hyp10 cells and DVA neurons (Supplementary Fig. 3).

2) The weak neuronal expression of ttr-11 could be masked by intestinal auto fluorescence. Use of an RFP variant would circumvent this problem. Otherwise, since an extrachromosomal construct was used, more than two animals should be analyzed to compensate for mosaic expression.

As suggested, we analyzed 10 animals (Supplementary Fig. 3).

3) Are the GFP-positive tail neurons a subset of the GABA neurons?

We identified the tail neurons expressing the *Pttr-11::nls::venus* reporter gene as the stretch sensitive sensory neurons DVA (Supplementary Fig. 3).

4) It is not clear why reduced MFG-E8-C2::GFP expression in ttr-11 mutants suggests that ttr-11 may localize around injured neurons. One might expect that TTR-11 would compete with MFG-E8-C2::GFP if they both bind to any PS that is present. If so, MGF-E8-C2::GFP levels would increase in ttr-11 mutants. This possibility should be addressed.

We examined the possibility that TTR-11 might compete with MFG-E8-C2::GFP to bind PS. However, overexpression of *ttr-11* did not decrease MFG-E8-C2::GFP expression levels after axon injury. This can be explained by the possibility that expression levels of MFG-E8-C2::GFP by the heat shock promoter may be higher than those of TTR-11.

During apoptosis, CED-7 mediates the release of PS from dying cells. In addition, TTR-52 could act as an extracellular PS carrier to facilitate the further movement of PS. We show that localization of MFG-E8-C2::GFP

around the injured neurons following axon injury was lower in the *ttr-11* mutant versus wild-type (Fig. 5c,d and Supplementary Fig. 8). This result suggests that TTR-11 also functions upstream to control PS accumulation after axon injury. It is therefore possible that TTR-11 is required for PS exposure by acting as an extracellular PS carrier that facilitates PS movement. We mention this possibility in “Results” (p. 13, line 318 ~ p. 14, line 321) and discuss it in “Discussion” (p. 10, line 466 ~ 473).

ttr-11 rescue:

1) The *Punc-25* experiments were done in young adults. Not using the same experimental paradigm as the rest of the study raises the question of why they were done that way. Was regeneration significantly different in young adults and L4 animals in any of the genotypes?

We show the *Punc-25* experiments done in L4 stage animals (Supplementary Fig. 2).

Regeneration was not different between young adults and L4 animals in any of the genotypes used in this work.

2) The data that *ttr-11* does not function cell autonomously (at least in part) are fairly weak and should be presented as such. For example, there is no significant difference in regeneration between wild type and *ttr-11*; *Punc-25::ttr-11* animals, suggesting *ttr-11* does function in GABA neurons.

We show that *Punc-25::ttr-11* failed to rescue the axon regeneration defect in *ttr-11* mutants (Supplementary Fig. 2 and Supplementary Table 1). D-type motor neurons may not have the component(s) required for TTR-11 function.

ttr-57:

RNAi of one (or both) mutant(s) in the background of the other deletion would address the question, and if negative, would strengthen the argument that binding to PS at Asn-46 is necessary for regeneration. Nonetheless, the text only states that the deletion has no effect on regeneration, so is accurate as written. At the least, the presence of *ttr-57* in this figure should be addressed in the text.

We used the CRIPSR/Cas9 system to introduce the *ttr-57(km85)* mutation into an animal having a *ttr-11(tm64)* background. We show that the *ttr-57* mutation did not enhance the defect in regeneration observed in *ttr-11(km64)* mutants (Supplementary Fig. 2 and Supplementary Table 1). Furthermore, we show that overexpression of *ttr-57* did not influence the *ttr-11* defect in regeneration (Supplementary Fig. 2 and Supplementary Table 1). These results indicate that TTR-57 is not involved in axon regeneration after laser axotomy.

Discussion of *ced-7* rescue (line 430):

it is possible that *ced-7* expression from neighboring GABA cells rescue regeneration in cut GABA neurons. Revise statement; CED-7 expressed in other than injured neuron is dispensable for axon regeneration.

As suggested, we have rewritten this part.

Our responses to the comments of Reviewer #2

Figure 1.

b. The authors could add representative of all, most or whatever, for clarity. To my experience this defect shown looks like best case scenario. The authors data in Figure 3 suggest that at best ~75% are scored as regenerating, so they need to be clear on that.

As described in “METHODS” (p. 27, line 700 ~ p. 28, line 701), proximal axon segments that showed no change after 24 hr were counted as “no regeneration”. Therefore, representative D-type motor neurons in *ttr-11* mutant animals after laser surgery were similar to that shown in Fig. 1b.

In wild-type animals, ~75% of cut axons were scored as regenerating. We mention this point in the legend of Fig. 1b. We have also rewritten the legend of Fig. 1b.

c. The legend should indicate D type neurons because of the odd result with of rescue using the touch neuron specific *mec-7* promoter that is included; one might confuse with touch neuron axotomies.

As suggested, we indicate D-type neurons in the legend of Fig. 1c.

Figure 2.

b. I wonder if the journal will request definition of abbreviations on panel.

We do not use abbreviations on the panel of Fig. 2b.

Data/story could be enhanced by using interaction-defective TTR-11 mutant.

To examine whether the TTR-11(N46A) mutation would affect its PS-binding activity statistically, we tried to purify GST-fused TTR-11 proteins from *E. coli*, but failed to do so due to protein insolubility.

c. Useful to define N46A as predicted interaction defective. Write N46A on panel.

As suggested, we have written N46A on the panel.

One wonders about the interaction test; IP INA first, then test TTR-11 FLAG.

We confirmed that immunoprecipitation of INA-1-ECD with an anti-GFP antibody co-immunoprecipitated TTR-11::FLAG, but co-immunoprecipitated TTR-11(N46A)::FLAG only weakly (Supplementary Fig. 7).

Figure 3.
Line 804 is diagram (singular).

As suggested, we have made this correction.

Figure 4.
a. Legend should indicate that mCherry is used to visualize the neuron.

As suggested, we indicate that mCherry is used to visualize the neuron in the legend of Fig. 4a.

b. Legend should indicate how long after axotomy scores were taken. Clarify the relative of what score to what score.

As suggested, we indicate how long after axotomy scores were taken in the legend of Fig. 4b.

As suggested, we describe the score formation in “METHODS” (p. 28, line 717 ~ 726).

c. panel is a bit hard to decipher at first glance. Possibly more clear if WT was replaced by MFG-E8-C2::GFP; don't the two WTs indicate different things?

As suggested, we have made this correction.

Would have been nice to repeat independently with Annexin V::GFP.

As suggested, we repeated this assay with Annexin V::GFP. We expressed Annexin V::GFP (AnxV::GFP) under the control of a heat-shock promoter. We show that axon injury induced the accumulation of AnxV::GFP signals around the injured neurons (Supplementary Fig. 9c).

****The biggest issue here is what is happening with the heat shock promoter; do the authors induce with hs, do they rely on the laser?**

We used the heat shock promoter to induce expression of MFG-E8-C2::GFP.

The accumulation of MFG-E8-C2::GFP around injured D-neurons is dependent on the laser.

Furthermore, we show that when MFG-E8-C2::GFP was expressed in D-type motor neurons by the *unc-25* promoter, D neuron axotomy induced MFG-E8-C2::GFP localization around the injured D neurons (Supplementary Fig. 9b).

Figure 5.
c. Do the authors specify uniform exposure for the collection of data in methods. How many are tested, how many is this representative of?

As suggested, we indicate that each image is representative of the 20-image series for each strain in the legend of Fig. 5c. All images were taken under the same conditions.

d. N numbers need to be added to the figure panel to indicate the rigor of the data. What is relative to what should be articulated in the legend.

As suggested, we indicate N numbers in the legend of Fig. 5d.

As suggested, we explain the relative score formation in “METHODS” (p. 28, line 717 ~ 726).

**Figure 7.
Needs much more explanation in the legend.**

As suggested, we present a more thorough explanation in the legend of Fig. 7.

****I think the model presentation has to be remade as it is misleading. There are no data that generally support that CED-3/CED-7 act in the distal fragment; most commonly, this most likely occurs in the injured neuron (though maybe in surrounding tissue). The authors could state that the current model is for the touch neuron potentiation of DA MN regeneration; but better to diagram the general model, with CED-3/CED-7 events on both sides, or just summarized on the side that regenerates.**

As suggested, we summarize CED-3/CED-7 events on the side that regenerates in Fig. 7.

Supplemental figures:

S Figure 2.

Title is misleading and should be changed. Data support expression in D Mneuron or touch neurons can complement regeneration defects; they do not on their own establish a non-autonomous mechanism. They show that the mechanism CAN be non-autonomous, but it can be autonomous.

We have changed the title of Supplementary Fig. 2.

We show that *Punc-25::ttr-11* failed to rescue the axon regeneration defect in *ttr-11* mutants (Supplementary Fig. 2 and Supplementary Table 1). D-type motor neurons may not have the component(s) required for TTR-11 function.

****S Figure 3.**

The expression study seems somewhat incomplete. This is a transcriptional fusion, regulatory and coding sequences may be missing. Can there be a repeat with the intact genomic version of the gene, even if functionality is not supported; is the pattern different?

As suggested, we also examined the expression pattern of *ttr-11* using the intact genomic version of the gene, *Pttr-11::ttr-11::gfp*. GFP expression was

still not observed around D-type motor neurons in L4 animals after axon injury (Supplementary Fig. 5)

****Looking at two animals is insufficient; what age is this animal, does the pattern change with age?**

We analyzed 10 animals (Supplementary Fig. 3).

In the revised experiments, we determined the expression pattern of the *ttr-11* gene in animals carrying *Pttr-11::nls::venus* and *Punc-25::nes::cfp*. CFP under the control of the *unc-25* promoter was used to visualize D neurons. We show that in L1 stage animals VENUS expression was observed in HSN neurons, excretory gland cells, hyp10 cells and DVA neurons (Supplementary Fig. 3). However, this expression pattern was not detected in the L4 stage animals.

****There are no data presented here regarding how the reporter responds to injury; information claimed on line 145. The science basis for this statement should be included.**

As suggested, we show that axon injury did not induce expression of *Pttr-11::nls::venus* in D neurons in the L4 stage animals (Supplementary Fig. 4).

**S Figure 5.
Indicate increase as compared to what.**

We clarify that the increase is over the mean GFP intensity in the ROI before axotomy. We explain this in “METHODS” (p. 28, line 727 ~ p. 29, line 735).

**S Figure 6.
Add how many animals were examined to show a similar time course. Surely this is not a single experiment.**

As suggested, we indicate that 5 animals were examined in the legend of revised Supplementary Fig. 9a.

**S Figure 8.
Can the authors indicate which D neurons are cut in these studies? Are they always the ones closest to the touch neuron? Do the more distant ones regenerate less efficiently than the ones close to the injured PLM?**

We have cut the VD9, DD5 and VD10 neurons. As suggested, we indicate this in the legend of revised Supplementary Fig. 11b.

D neurons have an intrinsically different regeneration ability that is not dependent on their distance from the PLM neuron. Therefore, it is difficult to conduct the experiment suggested by this reviewer.

S Table 2.

Some of the nomenclature designations are unclear. Each Ex transgene should be spelled out for genotype in this list.

As suggested, we have spelled out each Ex transgene for genotype in this list.

Text.

Line 84.

Authors might consider better phrasing: ...does not have a MFG-E8 homolog easily detected by sequence homology. My point is that there can be functional homologs, which is part of the point of this paper.

As suggested, we have rewritten this part.

Line 101.

The authors might be just a bit more careful in stating the conclusion; there is reasonable evidence for this model; but the precise signaling relationship is not proven and could involve other players, not necessarily in direct interaction. How about; Our data support a model in which TTR-11---

As suggested, we have rewritten this part.

Line 121.

the figure 1a referenced does not show the reduced expression, so the statement is misleading. Is the expression data in the paper? It could be refer to it, as supplementary. But I think the intention can be met by changing the statement: svh-13 RNAi is expected to target both *ttr-11* and *ttr-57* genes.

As suggested, we have rewritten this part.

****Around line 141.**

As noted above, the expression data as documented are not compelling. The authors site outcome based on two animals, which is not enough. The paper might be enhanced by adding on a full length clone expression and localization, even in the absence of demonstrated functionality. The authors have this construct in hand. One wonders whether testing in single copy might result in functionality, given concerns about potential dominant negative effects when a certain level of overexpression is attained.

We analyzed 10 animals (Supplementary Fig. 3).

As suggested, we examined the expression pattern of *ttr-11* using the intact genomic version of the gene, *Pttr-11::ttr-11::gfp*. GFP expression was still not observed around D neurons in L4 stage animals after axon injury (Supplementary Fig. 5).

We confirmed that the *Pttr-11::ttr-11::gfp* transgene did not show a dominant negative effect on axon regeneration. Fusion of GFP to either the C-terminus or the N-terminus after the signal sequence of *ttr-11* may interfere with its function.

****The authors state no change in this construct with injury; where is that data, what are the N numbers there? The legend for S Figure 3 does not show this. A key interesting question is where the TTR-11 moves upon injury; great if that could be addressed.**

We analyzed 5 animals (Supplementary Fig. 4).

We failed to detect VENUS or GFP signals in or around D neurons from animals carrying *Pttr-11::nls::venus* or *Pttr-11::ttr-11::gfp*, respectively, in L4 stage animals after axon regeneration (Supplementary Fig. 4 and 5).

During apoptosis, CED-7 mediates the release of PS from dying cells. In addition, TTR-52 could act as an extracellular PS carrier to facilitate the further movement of PS. We show that localization of MFG-E8-C2::GFP around the injured neurons following axon injury was lower in the *ttr-11* mutant versus wild-type (Fig. 5c,d and Supplementary Fig. 8). This result suggests that TTR-11 is required for PS exposure by acting as an extracellular PS carrier that facilitates PS movement. Therefore, it is likely that TTR-11 would move to injured neurons after axon injury.

Another thing that the authors should consider here is the question of gene dosage in these rescues. Virtually all their studies are over-expression construct, and the level of expression may differ for each transgene array. The MFG-E8-C2 can have a dominant negative effect, presumably by binding PS or obstructing access to other interactors; wouldn't the same be potentially expected for elevated TTR-11? Only single lines are reported. The authors should include some mention of this caveat for interpretation of rescue studies in some legend or supplementary site.

As suggested, we mention this caveat for the interpretation of rescue studies in "METHODS" (p. 27, line 689 ~ 691).

Line 154.

Better to state that TTR-11 can act non-autonomously for axonal regeneration.

As suggested, we have made this correction.

Line 188.

Better to state that TTR-11 can act as.

As suggested, we have made this correction.

****Page 10, experiments on *mfg-e8-c2::gfp*. These studies are really cool. Still, the use of the heat shock promoter introduces some potential complications; was heat shock used to induce expression? What is the impact of heat shock on the regeneration response? There is no mention of how the study was conducted with regard to any potential induction, which has to be made clear.**

We used the heat shock promoter to induce expression of MFG-E8-C2::GFP.

The accumulation of MFG-E8-C2::GFP around injured D-neurons is dependent on the laser.

Furthermore, we show that when MFG-E8-C2::GFP was expressed in D-type motor neurons by the *unc-25* promoter, D neuron axotomy induced the localization of MFG-E8-C2::GFP around the injured D neurons (Supplementary Fig. 9b).

We have described the heat shock condition used to induce MFG-E8-C2::GFP in “METHODS” (p. 28, line 707 ~ 709).

Under this heat shock condition, the frequencies of axon regeneration after axon injury decreased. However, expression of wild-type MFG-E8-C2::GFP, but not MFG-E8-C2(AAA)::GFP, inhibited axon regeneration more strongly (Fig. 4c).

Line 370.

Need to restate this sentence as really the data shown regard calcium change/regulation in regard to PS deposition. Summarize that PS depends on upstream or parallel ca signal; then in the following sentence state the whole model.

As suggested, we have rewritten this part.

Discussion.

--need to articulate somewhere that this is only one part of the regeneration story; all PS signal is not lost in *ced-7* or other backgrounds, all regeneration is not lost either, authors should explicitly state this.

As suggested, we have stated this point.

line 464.

use secreted rather than secretory.

As suggested, we have made this correction.

line 467.

not clear why TTR-11 N46A should reduce PS binding without more data. PS should also bind TTR-11. Could the TTR-11 be tested as was done for INA-1 for PS binding in the biochemical assay, and use N46A as a control?

We show that TTR-11 associated with the extracellular domain of INA-1 in vitro (Fig. 2c and Supplementary Fig. 7). This suggests that PS binding is not necessary for the association of TTR-11 with INA-1. Therefore, we have deleted this part in “Discussion”.

Furthermore, we show that the in vitro association between the TTR-11(N46A) mutated form and INA-1 was significantly weaker (Fig. 2c

and Supplementary Fig. 7). This result suggests that the Asn-46 site in TTR-11 is important for binding to INA-1. We have rewritten the part of the interaction of TTR-11 with PS and INA-1 (p. 8, line 160 ~ p. 9, line 194).

REVIEWERS' COMMENTS:

Reviewer #1 (Remarks to the Author):

The authors have addressed my concerns, the conclusions support the presented data and the paper is clearly written. In my opinion, it is suitable for publication in Nature Communications.

Reviewer #2 (Remarks to the Author):

NCOMMS-17-19090A Hisamoto et al.

The authors have revised according to previous review. Key improvements are adding data that makes rigor in experimentation readily accessible, adds more extensive ttr-1 expression analysis, and adds controls on PS detection. The paper is interesting and advances understanding--as such it is suitable for publication at Nature Communications, although I would suggest that the authors consider the following:

The expression pattern for ttr-1 remains a bit of a puzzle as Pttr-1ttr-1-gfp appears not to be highly expressed in L4 or late adult even consequent to injury; this lack of signal might be best explained by normally poor GFP signals for proteins once they are secreted? It would have been nice for the authors to show this construct does not rescue--to fit their idea about the tag disrupting function; better yet try the tag somewhere else or add a transmembrane domain or remove the signal sequence, but I think they have done a lot and tying this experimental knot is not necessarily critical.

The authors show ced-10 gf expressed in GABA neurons from the unc-25 promoter can bypass the need for ced-7. This is a "cell autonomous" effect, as GABA neurons are the ones subjected to axotomy. In the worm literature, ced-10 acts in the "engulfing" cell rather than the dying cell-to mediate corpse phagocytosis, which is somewhat of a non-autonomous mechanism. I think the authors should add a few comments to note this and explain what they think might be going on for this ced-10 "autonomous" activity.

Our responses to the comments of Reviewer #2

The expression pattern for *ttr-1* remains a bit of a puzzle as *Pttr-1ttr-1-gfp* appears not to be highly expressed in L4 or late adult even consequent to injury; this lack of signal might be best explained by normally poor GFP signals for proteins once they are secreted? It would have been nice for the authors to show this construct does not rescue-to fit their idea about the tag disrupting function; better yet try the tag somewhere else or add a transmembrane domain or remove the signal sequence, but I think they have done a lot and tying this experimental knot is not necessarily critical.

We tried to construct a fusion of GFP to either the C-terminus or the N-terminus after the signal sequence of the *ttr-11* gene. However, neither produced a functional fusion gene. We mention this in the Result section.

The authors show *ced-10* gf expressed in GABA neurons from the *unc-25* promoter can bypass the need for *ced-7*. This is a "cell autonomous" effect, as GABA neurons are the ones subjected to axotomy. In the worm literature, *ced-10* acts in the "engulfing" cell rather than the dying cell-to mediate corpse phagocytosis, which is somewhat of a non-autonomous mechanism. I think the authors should add a few comments to note this and explain what they think might be going on for this *ced-10* "autonomous" activity.

As suggested, we discussed this point in the Discussion section.